# ChunkKV: Semantic-Preserving KV Cache Compression for Efficient Long-Context LLM Inference

## Abstract

Large Language Models (LLMs) have demonstrated remarkable capabilities in processing extensive contexts, but this ability comes with significant GPU memory costs, particularly in the key-value (KV) cache. Although recent KV cache compression methods show strong performance, all use discrete tokens to maintain the KV cache, leading to a loss of chunk semantic information. We introduce ChunkKV, a novel KV cache compression method that retains the most informative semantic chunks while discarding the less important ones. ChunkKV preserves semantic information by grouping related tokens. Furthermore, ChunkKV exhibits a higher similarity in the indices of the retained KV cache across different layers, so we also propose a layer-wise index reuse technique to further reduce computational overhead. This technique not only improves compression efficiency, but also provides insight into the similarities between layers within LLMs. We evaluated ChunkKV on long-context benchmarks including LongBench and Needle-In-A-HayStack, as well as the GSM8K in-context learning benchmark. Our experiments, conducted with models LLaMA-3-8B-Instruct, Mistral-7B-Instruct, and Qwen2-7B-Instruct, demonstrate that ChunkKV outperforms other KV cache compression methods in performance, even surpassing the full KV cache under the same conditions. With a compression ratio of 10%, ChunkKV achieves state-of-the-art performance on various tasks, indicating its effectiveness in semantic preservation and model performance for long-context and in-context LLM inference.

## 1 Introduction

Large Language Models (LLMs) have become essential for addressing various downstream tasks of natural language processing (NLP), including summarization and question answering, which require the interpretation of a wide context from sources such as books, reports, and documents, often encompassing tens of thousands of tokens (Raffel et al., 2020; Brown et al., 2020; Chowdhery et al., 2022; Tay et al., 2022; Touvron et al., 2023a;b). Recent advances in long-context technology within the field of machine learning (ML) systems (Dao et al., 2022; Dao, 2024; Jacobs et al., 2023; Xiao et al., 2024) and model architecture design (Chen et al., 2023a; Xiong et al., 2024; Chen et al., 2023b; Peng et al., 2024) have significantly enhanced the ability of LLMs to process increasingly large input context lengths (Liu et al., 2024b; Young et al., 2024), such as the Gemini-1.5-pro model, which can manage documents up to 1,500 pages in length (Reid et al., 2024). However, this ability to handle long contexts also presents significant challenges regarding the key-value (KV) cache for super-long prompts. For instance, the KV cache for a single token in a 7 billion-parameter model requires approximately 0.5 MB of GPU memory, resulting in a 10,000-token prompt consuming around 5 GB of GPU memory, which constitutes nearly one fifth of the memory available on an RTX 4090 GPU. Larger contexts will further increase GPU memory consumption during inference serving (AI21, 2024; X.AI, 2024; Reid et al., 2024; Anthropic, 2024b; DeepSeek-AI, 2024). Consequently, KV cache compression methods have become crucial technologies for reducing GPU memory costs when deploying LLM services.

To address the substantial GPU memory consumption caused by KV caching, recent studies have explored various optimization techniques. An effective approach involves compressing the KV cache

Figure 1: Illustration of the impact of the token discrete method and the chunk method on semantic preservation. The discrete method preserves words related to the question but often omits the subject. In contrast, the chunk method retains the subject of the words, maintaining more accurate semantic information. For the equation: $S$ is the score function, and $c$ is a chunk of tokens.

by pruning non-important discrete parts from the prompt tokens (Xiao et al., 2024; Zhang et al., 2023; Li et al., 2024; Ge et al., 2023; Zhang et al., 2024b; Fu et al., 2024; Yang et al., 2024b; Adnan et al., 2024; Liu et al., 2023; Tang et al., 2024; Fu et al., 2024). H2O (Zhang et al., 2023) and SnapKV (Li et al., 2024) have shown that retaining less than 50% of the discrete KV cache can significantly reduce GPU memory usage with minimal impact on performance. However, previous methods mainly focus on discrete token compression, which may result in the loss of semantic information. Although SnapKV apply pooling strategy, it still cannot preserve the semantic information. Figure 1 shows an example in which the high-sparsity discrete method preserves the words related to the question but often omits the subject, leading to a potential misinterpretation of the context. For example, in a passage discussing other animals that eat strawberries, the discrete method might erroneously retain the word "strawberries" while omitting crucial information about the subjects (i.e., other animals). This selective preservation can result in the loss of essential semantic information and can potentially lead to incorrect inferences or responses from the model. Such issues are particularly pronounced in multi-document QA tasks, where maintaining context across multiple sources is crucial for accurate comprehension and response generation. For more details on discrete token methods, please refer to APPENDIX A.

Table 1: Comparison of Methods on KV Cache Compression.

| Method | KV Cache Compression | Dynamic Policy | Layer-Wise Policy | Semantic Information | Efficient Index Reuse |
|---|:---:|:---:|:---:|:---:|:---:|
| StreamingLLM (Xiao et al., 2024) | ✓ | | | | |
| H2O (Zhang et al., 2023) | ✓ | ✓ | | | |
| SnapKV (Li et al., 2024) | ✓ | ✓ | | | |
| PyramidInfer (Yang et al., 2024b) | ✓ | ✓ | ✓ | | |
| PyramidKV (Zhang et al., 2024b) | ✓ | ✓ | ✓ | | |
| ChunkKV(Ours) | ✓ | ✓ | ✓ | ✓ | ✓ |

To address this gap, we explore the semantic dimensions of KV cache compression. We introduce a straightforward yet effective method, **ChunkKV**, which retains the most informative **semantic chunks** from the original KV cache, as in Figure 1. As outlined in Table 1, recent highly relevant KV cache compression methods *lack the ability to retain semantic information and efficiently reuse indices*. Furthermore, we investigate that *the preserved KV cache indices by ChunkKV exhibit a higher similarity* compared to previous methods. Consequently, we develop a technique called layer-wise index reuse, which reduces the additional computational time introduced by the KV cache compression method. To evaluate the performance of ChunkKV, we conduct experiments on long-context benchmarks, including LongBench (Bai et al., 2024) and Needle-In-A-HayStack (NIAH)(Kamradt, 2023), as well as in-context learning benchmarks such as GSM8K(Cobbe et al., 2021). Our ex-

periments demonstrate that ChunkKV outperforms other KV cache compression methods in both efficiency and accuracy, showing that retaining chunks of the original KV cache preserves more essential information. This indicates that ChunkKV is a simple yet effective method of compressing the KV cache.

We summarize our key contributions as follows:

- We identify the phenomenon in which discrete KV cache compression methods inadvertently prune the necessary semantic information.
- We propose ChunkKV, a simple KV cache compression method that uses the fragmentation method that keeps the semantic information and achieves state-of-the-art performance on long-context benchmarks.
- We propose the layer-wise index reuse technique to reduce the additional computational time introduced by the KV caching method.

## 2  RELATED WORK

**KV Cache Compression**    KV cache compression technology has developed rapidly in the era of LLM, with methods mainly focused on evicting unimportant tokens. The compression process occurs before the attention blocks, optimizing both the prefilling time and GPU memory. Xiao et al. (2024) and Han et al. (2024) propose that initial and recent tokens consistently have high attention scores between different layers and attention heads. As a result, retaining these tokens in the KV cache is more likely to preserve important information. Furthermore, FastGen (Ge et al., 2023) evicts tokens based on observed patterns. H2O (Zhang et al., 2023) and SnapKV (Li et al., 2024) employ dynamic KV cache compression methods, evaluating the importance of tokens based on attention scores and then evicting the less important ones. As inference scenarios become increasingly complex, dynamic KV cache compression methods demonstrate powerful performance. Recently, Yang et al. (2024b) and Zhang et al. (2024b) have closely examined the distributions of attention scores during the pre-filling stage of the Retrieval-Augmented Generation (RAG) task, discovering a pyramidal KV cache compression pattern in different transformer layers.

Although these KV cache compression methods have explored efficient GPU memory management while maintaining original performance, our study focuses more on the semantic information of the prompt. We find that chunks of the original KV cache are more important than discrete tokens.

**Chunking Method**    The chunking methodology is widely used in the field of NLP due to its simplicity and effectiveness (Tjong Kim Sang & Veenstra, 1999). In the era of LLMs, chunking is primarily applied in data pre-processing. For example, Shi et al. suggest grouping related training data into chunks to achieve better convergence curves to pre-train LLMs. Fei et al. (2024) apply a topic-based chunking method to improve the semantic compression of prompts. Furthermore, chunking plays an important role in the Retrieval-Augmented Generation (RAG) field (Yepes et al., 2024; Smith & Troynikov, 2024; Anthropic, 2024a). It serves to divide documents into units of information with semantic content suitable for embedding-based retrieval and processing by LLMs.

**Layer-Wise Technique**    The layer-wise technique is widely used in the training and inference of large language models (LLMs). LISA (Pan et al., 2024a) is a layer-wise sampling method based on observations of the training dynamics of Low-Rank Adaptation (LoRA)(Hu et al., 2022) across layers. LAMB(You et al., 2020) is a layer-wise adaptive learning rate method that speeds up LLM training by stabilizing training convergence with large batch sizes. DoLa (Chuang et al., 2023) employs layer-wise contrasting to reduce hallucinations during LLM inference.

## 3  CHUNKKV

### 3.1  PRELIMINARY STUDY OF KV CACHE COMPRESSION

With the increasing long-context capabilities of LLMs, the KV cache has become crucial for improving inference efficiency. However, it can consume significant GPU memory when handling

long input contexts. The GPU memory cost of the KV cache for the decoding stage can be calculated as follows:

$$\text{GPU Memory Cost of KV Cache} = 2 \times B \times S \times L \times N \times D \times 2 \tag{1}$$

where $B$ is the batch size, $S$ is the sequence length of prompt and decoded length, $L$ is the number of layers, $N$ is the number of attention heads, $D$ is the dimension of each attention head, and the first 2 accounts for the KV matrices, while the last 2 accounts for the precision when using float16. Table 2 shows the configuration parameters for LLaMA-3-8B-Instruct (Meta, 2024). With a batch size $B = 1$ and a sequence length of prompt $S = 2048$, the GPU memory cost of the KV cache is nearly 1 GB. If the batch size exceeds 24, the GPU memory cost of the KV cache will exceed the capacity of an RTX 4090 GPU. To address this issue, KV cache compression methods have been proposed, with the aim of retaining only a minimal amount of KV cache while preserving as much information as possible. For more details on the LLM configuration parameters, refer to APPENDIX D.

To optimize memory usage, a strategy called KV cache compression has been proposed Zhang et al. (2023); Xiao et al. (2024); Li et al. (2024). This strategy involves retaining only a minimal amount of KV cache while preserving as much information as possible, effectively reducing $L$ in Equation 1. Typically, the $L$ after applying KV compression methods is less than 50% of the original $L$, with minimal performance degradation. However, these methods primarily focus on discrete tokens of the KV cache, which may result in the loss of semantic information.

Table 2: LLaMA-3-8B-Inst Configuration Parameters.

| Attribute | Value |
|---|---|
| Model Name | LLaMA-3-8B-Inst |
| $L$ (Number of layers) | 32 |
| $N$ (Number of attention heads) | 32 |
| $D$ (Dimension of each head) | 128 |

### 3.2 PROPOSED METHOD

#### 3.2.1 CHUNKKV

To address the limitations of existing KV cache compression methods, we propose ChunkKV, a novel KV cache compression method that retains the most informative semantic chunks. The key idea behind ChunkKV is to group tokens in the KV cache into chunks that preserve more semantic information, such as a chunk containing a subject, verb and object. As illustrated in Figure 1, ChunkKV preserves the chunks of the KV cache that contain more semantic information. First, we define a chunk as a group of tokens that contain related semantic information. By retaining the most informative chunks from the original KV cache, ChunkKV can effectively reduce the memory usage of the KV cache while preserving essential information.

The Algorithm 1 shows the pseudocode outline of ChunkKV. First, following H2O (Zhang et al., 2023) and SnapKV (Li et al., 2024), we set the observe window by computing the observation scores $\mathbf{A} \leftarrow \mathbf{Q}_{T_q-w:T_q}\mathbf{K}^T$, where $\mathbf{Q}_{T_q-w:T_q}$ is the observe window, $\mathbf{K}$ is the Key matrix and the window size $w$ is usually set to $\{4, 8, 16, 32\}$. Next, the number of chunks $C$ is calculated as $C = \lceil \frac{T_k}{c} \rceil$, where $T_k$ is the length of the Key matrix and $c$ is the chunk size. The observation scores for each chunk are then computed as $\mathbf{A}_i = \sum_{j=(i-1)c+1}^{ic} \mathbf{A}_{:,j}$ for $i = 1, 2, \ldots, C$. Referring to previous works (Zhang et al., 2023; Li et al., 2024; Yang et al., 2024b; Zhang et al., 2024b), we still use the top-$k$ algorithm as ChunkKV's sampling policy. For the top-$k$ chunk selection, the top-$k$ chunks are selected based on their observation scores, where $k = \lfloor \frac{L_{\max}}{c} \rfloor$, and $L_{\max}$ is the maximum length of the compressed KV cache. The size of the last chunk will equal $\min(c, L_{\max} - (k-1) \times c)$. The indices of the top-$k$ chunks will keep the original sequence order. In the compression step, the key and value matrices are only retained based on the selected indices, resulting in the compressed KV cache. Finally, the observe window of the original KV cache will be concatenated to the compressed KV cache by replacing the last $w$ tokens to keep important information. The compressed KV cache is then used for subsequent attention computations.

#### 3.2.2 LAYER-WISE INDEX REUSE

---

**Algorithm 1** ChunkKV: Algorithm for KV Cache Compression

---

1: **Input:** $\mathbf{Q} \in \mathbb{R}^{T_q \times d}$, $\mathbf{K} \in \mathbb{R}^{T_k \times d}$, $\mathbf{V} \in \mathbb{R}^{T_v \times d}$, observe window size $w$, chunk size $c$, compressed KV cache max length $L_{\max}$
2: **Output:** Compressed KV cache $\mathbf{K}'$, $\mathbf{V}'$
3: **Observe Window Calculation:**
4: $\mathbf{A} \leftarrow \mathbf{Q}_{T_q-w:T_q} \mathbf{K}^T$         ▷ Attention scores for the observe window
5: $C \leftarrow \left\lceil \frac{T_k}{c} \right\rceil$            ▷ Calculate the number of chunks
6: **Chunk Attention Score Calculation:**
7: **for** $i = 1$ to $C$ **do**
8:    $\mathbf{A}_i \leftarrow \sum_{j=(i-1)c+1}^{ic} \mathbf{A}_{:,j}$      ▷ Sum of observation scores for each chunk
9: **end for**
10: **Top-K Chunk Selection:**
11: $k \leftarrow \left\lfloor \frac{L_{\max}}{c} \right\rfloor$
12: *Top_K_Indices* $\leftarrow$ indices of Top-$k$ chunks based on $\mathbf{A}_i$
13: **Compression:**
14: $\mathbf{K}', \mathbf{V}' \leftarrow$ index_select($\mathbf{K}, \mathbf{V}, \textit{Top\_K\_Indices}$)
15: **Concatenation:**
16: $\mathbf{K}' \leftarrow$ concat($\mathbf{K}'_{0:L_{\max}-w}, \mathbf{K}_{T_k-w:T_k}$)
17: $\mathbf{V}' \leftarrow$ concat($\mathbf{V}'_{0:L_{\max}-w}, \mathbf{V}_{T_v-w:T_v}$)
18: **return** $\mathbf{K}'$, $\mathbf{V}'$

---

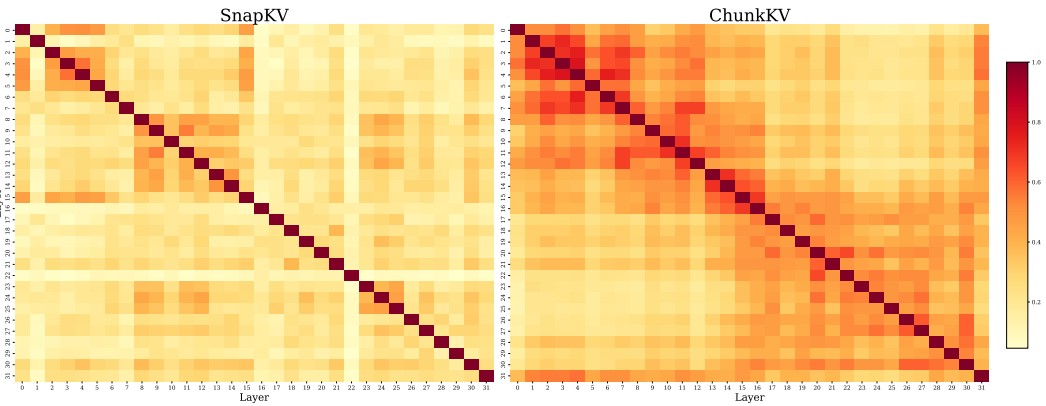

Figure 2: Layer-wise similarity heatmaps of the preserved KV cache indices by SnapKV (left) and ChunkKV (right) on LLaMA-3-8B-Instruct.

Furthermore, we investigated the preserved KV cache indices by ChunkKV and found that they exhibit higher similarity compared to previous methods. Figure 2 shows the layer-wise similarity heatmaps of SnapKV and ChunkKV. Each cell represents the similarity between the preserved KV cache indices of two layers, with deeper colors indicating higher similarity. The results demonstrate that the KV cache indices preserved by ChunkKV are more similar to those in neighboring layers. As shown in Table 3, ChunkKV consistently achieves a higher average Jaccard similarity between adjacent layers compared to SnapKV in different model architectures, indicating that the retained token index in ChunkKV is more similar to each other. For a more detailed visualization, please refer to Appendix B.2.1.

**Algorithm 2** Layer-wise Index Reuse for ChunkKV

---

1: **Input:** Number of layers in LLMs $N_{\text{layers}}$, number of reuse layers $N_{\text{reuse}}$
2: **Initialize:** Dictionary to store indices $\mathcal{I}_{\text{reuse}} = \{\}$
3: **for** $l = 0$ to $(N_{\text{layers}} - 1)$ **do**
4:    **if** $l \mod N_{\text{reuse}} == 0$ **then**
5:      $\mathbf{K}'_l, \mathbf{V}'_l, \mathcal{I}_l \leftarrow$ ChunkKV($\mathbf{K}_l, \mathbf{V}_l$)
6:      $\mathcal{I}_{\text{reuse}}[l] \leftarrow \mathcal{I}_l$
7:    **else**
8:      $\mathcal{I}_l \leftarrow \mathcal{I}_{\text{reuse}}\left[\left\lfloor \frac{l}{N_{\text{reuse}}} \right\rfloor \times N_{\text{reuse}}\right]$
9:    **end if**
10:    $\mathbf{K}'_l \leftarrow$ index_select($\mathbf{K}_l, \mathcal{I}_l$)
11:    $\mathbf{V}'_l \leftarrow$ index_select($\mathbf{V}_l, \mathcal{I}_l$)
12: **end for**

Based on the above findings, we propose a training-free *layer-wise index reuse* method to further reduce the additional cost of the KV cache compression time, which reuses compressed token indices across multiple layers. This procedure is formally described in Algorithm 2. The ChunkKV compression process returns the compressed KV cache and their respective token indices, denoted as $\mathcal{I}_l$. For layer-wise index reuse, we define a grouping of layers such that all $N_{\text{reuse}}$ layers share the same token in-

Table 3: Retained KV Cache Indices Similarity of Adjacent Layers for Different Models.

| Method | H2O | SnapKV | ChunkKV |
|--------|-----|--------|---------|
| LLaMA-3-8B | 25.31% | 27.95% | **57.74%** |
| Qwen2-7B | 14.91% | 16.50% | **44.26%** |
| Mistral-7B | 15.15% | 15.78% | **52.16%** |

dices for ChunkKV. Specifically, for a group of layers $\{l, l+1, \ldots, l + N_{\text{reuse}} - 1\}$, we perform ChunkKV on the first layer $l$ to obtain the token indices $\mathcal{I}_l$ and reuse $\mathcal{I}_l$ for the subsequent layers $l + 1, l + 2, \ldots, l + N_{\text{reuse}} - 1$. The notation $\mathbf{K}_l[\mathcal{I}_l]$ and $\mathbf{V}_l[\mathcal{I}_l]$ indicates the selection of key and value caches based on the indices in $\mathcal{I}_l$.

**Efficiency Analysis** The layer-wise index reuse method significantly reduces the computational complexity of ChunkKV. Without index reuse, ChunkKV would be applied to all $N_{\text{layers}}$ layers, resulting in a total compression time of $N_{\text{layers}} \cdot T_{\text{compress}}$, where $T_{\text{compress}}$ is the time taken to compress one layer. With index reuse, ChunkKV is only applied to $\frac{N_{\text{layers}}}{N_{\text{reuse}}}$ layers, reducing the total time to $\frac{N_{\text{layers}}}{N_{\text{reuse}}} \cdot T_{\text{compress}} + (N_{\text{layers}} - \frac{N_{\text{layers}}}{N_{\text{reuse}}}) \cdot T_{\text{select}}$, where $T_{\text{select}}$ is the time taken to select indices, which is typically much smaller than $T_{\text{compress}}$. This results in a theoretical speedup factor of:

$$\text{Speedup} = \frac{N_{\text{layers}} \cdot T_{\text{compress}}}{\frac{N_{\text{layers}}}{N_{\text{reuse}}} \cdot T_{\text{compress}} + (N_{\text{layers}} - \frac{N_{\text{layers}}}{N_{\text{reuse}}}) \cdot T_{\text{select}}}$$

Assuming $T_{\text{select}}$ is negligible compared to $T_{\text{compress}}$, this simplifies to approximately $N_{\text{reuse}}$. In practice, the actual speedup may vary depending on the specific implementation and hardware, but it can still lead to substantial time savings, especially for models with a large number of layers. For more details, please refer to Appendix B.1.

## 4 EXPERIMENT RESULTS

In this section, we conduct experiments to evaluate the effectiveness of ChunkKV on KV cache compression in two benchmark fields, with a chunk size set to 10 even for various model architectures. The first is the Long-Context benchmark, which includes LongBench (Bai et al., 2024) and Needle-In-A-HayStack (NIAH) (Kamradt, 2023), both widely used for assessing KV cache compression methods. The second is the In-Context Learning benchmark, for which we select GSM8K (Cobbe et al., 2021) to evaluate the performance of ChunkKV. The In-Context Learning scenario is a crucial capability for LLMs and has been adapted in many powerful technologies such as Chain-of-Thought (Wei et al., 2022; Diao et al., 2024; Pan et al., 2024b). GSM8K is widely used to evaluate In-Context Learning methods and contains more than 1,000 arithmetic questions. All experiments were conducted three times, using the mean score to ensure robustness.

### 4.1 LONG-CONTEXT BENCHMARK

LongBench and NIAH are two widely used benchmarks for KV cache compression methods. Both benchmarks have a context length that exceeds $10K$. NIAH requires retrieval capability, while LongBench is a meticulously designed benchmark suite that tests the capabilities of language models in handling extended documents and complex information sequences.

#### 4.1.1 LONGBENCH

**Settings** We use LongBench (Bai et al., 2024) to assess the performance of ChunkKV on tasks involving long-context inputs. LongBench is a meticulously designed benchmark suite that evaluates the capabilities of language models in handling extended documents and complex information sequences. This benchmark was created for multi-task evaluation of long-context inputs and includes 17 datasets covering tasks such as single-document QA (Kočiský et al., 2018; Dasigi et al., 2021),

multi-document QA (Yang et al., 2018; Ho et al., 2020; Trivedi et al., 2022; He et al., 2018), summarization (Huang et al., 2021; Zhong et al., 2021; Fabbri et al., 2019; Wu et al., 2023), few-shot learning (Li & Roth, 2002; Gliwa et al., 2019; Joshi et al., 2017), synthetic tasks and code generation (Guo et al., 2023; Liu et al., 2024d). The datasets feature an average input length ranging from $1K$ to $18K$ tokens, requiring substantial memory for KV cache management. For more details on LongBench, please refer to the APPENDIX E. We evaluated multiple KV cache eviction methods using the LongBench benchmark with LLaMA-3-8B-Instruct (Meta, 2024), Mistral-7B-Instruct-v0.3 (Jiang et al., 2023a), and Qwen2-7B-Instruct (Yang et al., 2024a), with a KV cache compression ratio of $10\%$. The LongBench provides the Chinese subtask, and Qwen2-7B-Instruct also supports Chinese, so we tested Qwen2-7B-Instruct with different KV cache compression methods on the Chinese subtask.

**Results** Tables 4 show that ChunkKV is capable of achieving on-par performance or even better than the full KV cache with less GPU memory consumption. This table is evaluated in the LongBench English subtask, where ChunkKV outperforms other compression methods overall, and the Qwen2-7B-Instruct model achieves better performance than the full KV cache. In particular, ChunkKV shows particularly strong performance in Multi-Document QA tasks, highlighting its ability to effectively preserve and utilize the context of the cross document. This suggests that ChunkKV's approach of retaining semantic chunks is more effective in preserving important information compared to other discrete token-based compression methods. For detailed results and Chinese subtask results, please refer to Appendix B.3 and B.6.

### 4.1.2 NEEDLE-IN-A-HAYSTACK

**Settings** We use Needle-In-A-HayStack (NIAH) (Kamradt, 2023) to evaluate LLMs' long-context retrieval capability. NIAH assesses how well LLM extract hidden tricked information from extensive documents, and follow LLM-as-a-Judge (Zheng et al., 2023) we apply GPT-4o-mini (OpenAI, 2023) to assess the accuracy of the retrieved information. We evaluated multiple KV cache eviction methods using NIAH with LLaMA-3-8B-Instruct and Mistral-7B-Instruct-v0.2, setting benchmark context lengths to 8k and 32k tokens.

**Results** Figure 3 presents the NIAH benchmark results for LLaMA-3-8B-Instruct. The vertical axis represents the depth percentage, while the horizontal axis represents the token length, with shorter lengths on the left and longer lengths on the right. A cell highlighted in green indicates that the method can retrieve the needle at that length and depth percentage. The detail visualization of the NIAH benchmark can be found in Appendix B.4. The visualization results demonstrate that ChunkKV outperforms other KV cache compression methods. Table 5

Table 5: NIAH Performance Comparison.

| Method | LLaMa-3-8B-Inst | Mistral-7B-Inst |
|---|---|---|
| StreamingLLM | 23.7 | 44.3 |
| H2O | 47.9 | 88.2 |
| SnapKV | 58.9 | 91.6 |
| PyramidKV | 65.1 | 99.3 |
| **ChunkKV** | **73.8** | **99.8** |
| FullKV | 74.6 | 99.8 |

provides statistical results for different compression methods. These findings clearly indicate the effectiveness of ChunkKV in managing varying token lengths and depth percentages, making it a robust choice for KV cache management in large language models.

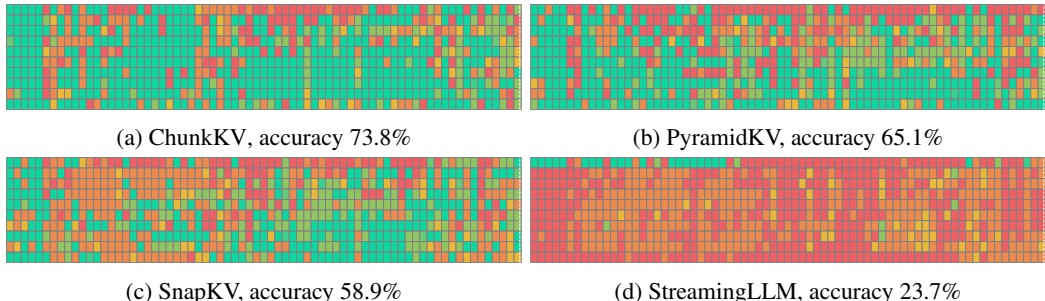

(a) ChunkKV, accuracy 73.8%  (b) PyramidKV, accuracy 65.1%

(c) SnapKV, accuracy 58.9%  (d) StreamingLLM, accuracy 23.7%

Figure 3: NIAH benchmark for LLaMA3-8B-Instruct with KV cache size=128 under 8k context length.

Table 4: Comparative analysis of KV cache compression methods on the LongBench English sub-task, including ChunkKV, PyramidKV, SnapKV, H2O, StreamingLLM, and FullKV. Results are shown for LLaMa-3-8B-Instruct, Qwen2-7B-Instruct, and Mistral-7B-Instruct models. ChunkKV demonstrates superior performance across diverse LLM architectures compared to other compression techniques.

| Method | Single-Document QA | Multi-Document QA | Summarization | Few-shot Learning | Synthetic & Code | Overall Avg. ↑ |
|---|---|---|---|---|---|---|
| Avg len | 8,862 | 8,417 | 7,154 | 6,548 | 6,468 | 7,490 |
| LlaMa-3-8B-Instruct, KV Size = Full | | | | | | |
| FullKV | 32.19 | 34.59 | 24.96 | 68.48 | 45.69 | 41.46 |
| LlaMa-3-8B-Instruct, KV Size Compression Ratio = 10% | | | | | | |
| StreamingLLM | 18.60 | 26.64 | 20.64 | 62.15 | 46.96 | 35.74 |
| H2O | 24.64 | 31.01 | 22.13 | 57.02 | 47.15 | 37.06 |
| SnapKV | **28.35** | 32.91 | **22.32** | 67.21 | 47.53 | 40.15 |
| PyramidKV | 27.40 | 32.76 | 22.60 | **67.88** | 47.38 | 40.08 |
| **ChunkKV** | 28.50 | **33.46** | 22.20 | 67.62 | **48.23** | **40.51** |
| LlaMa-3-8B-Instruct, KV Size Compression Ratio = 20% | | | | | | |
| StreamingLLM | 25.09 | 30.17 | **24.29** | 66.89 | 45.40 | 38.80 |
| H2O | 27.61 | 31.49 | 23.01 | 59.25 | 45.17 | 37.79 |
| SnapKV | 29.96 | 33.33 | 23.91 | 67.98 | 45.74 | 40.53 |
| PyramidKV | 30.08 | **33.76** | 24.14 | **68.00** | 45.56 | 40.63 |
| **ChunkKV** | **31.05** | 33.22 | 23.58 | 67.86 | **46.19** | **40.74** |
| LlaMa-3-8B-Instruct, KV Size Compression Ratio = 30% | | | | | | |
| StreamingLLM | 27.43 | 32.04 | **25.08** | 68.03 | 47.48 | 40.48 |
| H2O | 28.65 | 32.77 | 23.76 | 61.07 | 47.23 | 39.23 |
| SnapKV | 31.06 | 33.76 | 24.34 | **68.40** | 47.55 | 41.43 |
| PyramidKV | 30.71 | **34.05** | 24.62 | 68.14 | 47.36 | 41.37 |
| **ChunkKV** | **31.48** | 33.26 | 24.53 | 68.34 | **48.15** | **41.59** |
| Mistral-7B-Instruct-v0.3, KV Size = Full | | | | | | |
| FullKV | 41.18 | 38.99 | 29.45 | 70.73 | 57.08 | 48.08 |
| Mistral-7B-Instruct-v0.3, KV Size Compression Ratio = 10% | | | | | | |
| StreamingLLM | 26.90 | 32.09 | 21.50 | 66.01 | 50.58 | 40.11 |
| H2O | 37.71 | 37.92 | 24.01 | 59.35 | 53.96 | 43.61 |
| SnapKV | 39.61 | 38.77 | **24.74** | 68.93 | **56.48** | 46.38 |
| PyramidKV | 39.25 | 39.08 | 25.12 | 70.03 | 55.73 | 46.39 |
| **ChunkKV** | **40.19** | **39.35** | 24.40 | **70.41** | 56.24 | **46.71** |
| Qwen2-7B-Instruct, KV Size = Full | | | | | | |
| FullKV | 37.35 | 12.18 | 28.78 | 70.62 | 51.17 | 40.71 |
| Qwen2-7B-Instruct, KV Size Compression Ratio = 10% | | | | | | |
| StreamingLLM | 37.34 | 11.44 | 26.84 | 70.40 | 44.36 | 38.56 |
| H2O | 37.68 | 11.47 | **27.29** | 70.09 | **51.92** | 40.45 |
| SnapKV | **39.53** | 12.51 | 27.05 | 70.30 | 50.18 | 40.55 |
| PyramidKV | 38.52 | 11.87 | 26.78 | 70.32 | 50.66 | 40.31 |
| **ChunkKV** | 38.57 | **13.02** | 27.05 | **70.50** | 51.74 | **40.88** |

## 4.2 IN-CONTEXT LEARNING

The In-Context Learning (ICL) ability significantly enhances the impact of prompts on large language models (LLMs). For example, the Chain-of-Thought approach (Wei et al., 2022) increases the accuracy of the GSM8K of the PaLM model (Chowdhery et al., 2022) from 18% to 57% without additional training. However, KV cache compression methods will potentially remove important prompt information. Therefore, evaluating KV cache compression methods using an ICL benchmark is an effective way to demonstrate the efficacy of different compression strategies.

### 4.2.1 GSM8K

**Settings** In the in-context learning scenario, we evaluated multiple KV cache compression methods for GSM8K (Cobbe et al., 2021), which contains more than 1,000 arithmetic questions on LLaMA-3-8B-Instruct (Meta, 2024) and Qwen2-7B-Instruct (Yang et al., 2024a). The KV cache compression ratio for this experiment is 30%. The CoT prompt settings for this experiment are the

same as those used by Wei et al. (2022). For more details on the prompt settings, please refer to the APPENDIX F.

**Results**   Table 6 presents the performance comparison. The results demonstrate that ChunkKV (CKV) outperforms other KV cache compression methods on both the LLaMa-3-8B-Instruct and Qwen2-7B-Instruct models. For LLaMa-3-8B-Instruct, ChunkKV achieves an accuracy of 74.6%, which is significantly higher than other compression methods and closely approaches the FullKV performance of 76.8%. This suggests that ChunkKV retains most of the important information needed for in-context learning, even with a compression ratio 30%. For

Table 6: GSM8K Performance Comparison.

| Method | LLaMa-3-8B | Qwen2-7B |
|---|---|---|
| StreamingLLM | 70.6 | 70.8 |
| H2O | 73.6 | 61.2 |
| SnapKV | 70.2 | 70.8 |
| PyramidKV | 68.2 | 64.7 |
| ChunkKV | **74.6** | **73.5** |
| FullKV | 76.8 | 71.1 |

Qwen2-7B-Instruct, ChunkKV not only outperforms other compression methods, but also surpasses the FullKV baseline, achieving an accuracy of 73.5% compared to FullKV's 71.1%. This improvement over the full KV cache indicates that ChunkKV's semantic-preserving approach may be particularly effective for certain model architectures, potentially filtering out noise and retaining the most relevant information for solving GSM8K problems. The consistent superior performance of ChunkKV in both models underscores its effectiveness in maintaining crucial contextual information for complex arithmetic reasoning tasks.

### 4.3   INDEX REUSE

This section will evaluate the performance of the layer-wise index reuse approach with ChunkKV from the two aspects of efficiency and performance.

#### 4.3.1   EFFICIENCY

**Settings**   This experiment evaluates the efficiency of the layer-wise index reuse approach by measuring the time cost of the KV cache compression method. We set the dummy prompt length to 50K tokens, and the chunk size to 10, with a KV cache compression ratio of 10%. Due to LLaMA3 not supporting long contexts, only Qwen2-7B-Instruct (Yang et al., 2024a) and Mistral-7B-Instruct-v0.3 (Jiang et al., 2023a) are used to evaluate the efficiency of the layer-wise index reuse approach.

**Results**   Figure 4 shows a clear trend towards increasing efficiency as the number of index reuse layers increases. For both models, the relative efficiency improves significantly with each increase in reuse layers, following a logarithmic pattern. Interestingly, the efficiency curves for both models are remarkably similar, suggesting that the benefits of layer-wise index reuse are consistent across different model architectures. The slight divergence at higher reuse layers may be attributed to specific architectural differences between different models. The logarithmic nature of the efficiency gains suggests that even a moderate number of reuse layers can yield substantial benefits, with diminishing returns at very high reuse levels.

#### 4.3.2   PERFORMANCE

**Settings**   This experiment evaluates the performance of the layer-wise index reuse approach by measuring the performance of the LongBench (Bai et al., 2024), the experiment settings are the

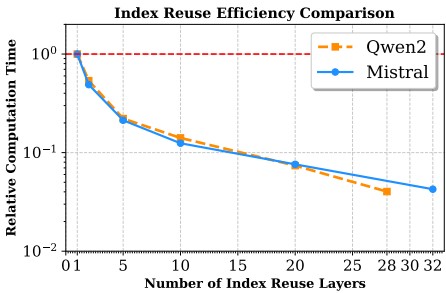

Figure 4: Relative computation time (lower is better) of layer-wise index reuse for KV cache compression in Qwen2 and Mistral models.

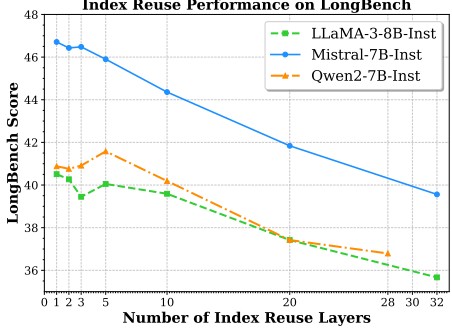

Figure 5: Comparison with different index reuse layers on LongBench.

same as section 4.1.1. And the number of index reuse layers is set from 1 to the number of layers in the model, where an index reuse layer of 1 corresponds to the normal ChunkKV without index reuse.

**Results** Figure 5 illustrates the performance of ChunkKV with varying index reuse layers on the LongBench benchmark. Generally, performance declines as the number of reuse layers increases, with the rate of decrease differing between models, but following a similar trend. In particular, the Qwen2-7B-Instruct model exhibits performance improvements 46.71% when the number of reuse layers ranges from 2 to 5. Table 7 presents the performance decrease rate from 0 layer index reuse to maximum layer index reuse on the LongBench benchmark, consistent with the values shown in Figure 5. The Qwen2 models demonstrate a lower performance decrease rate compared to LLaMA3 and Mistral. For more experiments on index reuse, please refer to the APPENDIX B.2.2.

Overall, these findings on efficiency and performance suggest that layer-wise index reuse can be an effective technique for optimizing the efficiency-performance trade-off in KV cache compression, with the potential for model-specific tuning to maximize benefits.

Table 7: Performance Degradation Rate of Layer-wise Index Reuse.

| Model | Performance Decrease Rate (%) |
|---|---|
| LLaMA-3-8B | 11.95 |
| Mistral-7B | 15.31 |
| Qwen2-7B | 10.00 |

## 5 ABLATION STUDY

### 5.1 CHUNK SIZE

This section aims to investigate the impact of chunk size on the performance of ChunkKV. Different chunk sizes will lead to varying degrees of compression on the semantic information of the data. We set the experiemnt setting the same as in section 4.1.1. The chunk size is set from the range $\{1, 3, 5, 10, 20, 30\}$. Figure 6 shows the performance of the ChunkKV with different chunk size on the LongBench benchmark. The three colorful curves represent three LLMs with different chunk sizes, and the colorful dashed line is the corresponding FullKV performance. For more experiments on the size of the chunks with different compression ratios, refer to the APPENDIX B.5.

From Figure 6, we can observe that the LongBench performance of ChunkKV is not significantly affected by the chunk size, with performance variations less than 1%. The three curves are closely aligned, indicating that chunk sizes in the range of $\{10, 20\}$ exhibit better performance. This finding aligns with the nature of semantic chunks in natural language, where different chunk sizes can retain different semantic information.

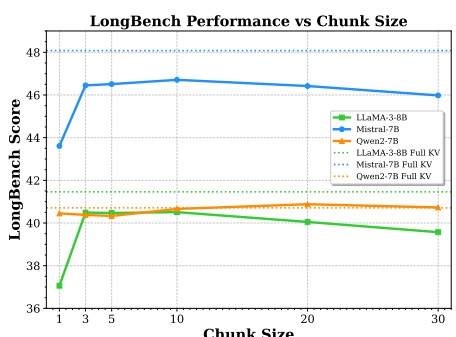

Figure 6: LongBench Performance Comparison with different chunk size.

## 6 CONCLUSION

We introduced ChunkKV, a novel KV cache compression method that preserves semantic information by retaining more informative chunks. Our extensive experiments demonstrate that ChunkKV consistently outperforms existing methods across various LLMs and benchmarks, often matching or surpassing full KV cache performance while using only a fraction of the memory. By focusing on semantic units rather than individual tokens, ChunkKV maintains crucial contextual information, leading to improved performance on complex tasks. Its effectiveness across different model architectures, languages, and chunk sizes highlights its versatility, while the proposed layer-wise index reuse technique offers extra speedup in the compression process with minimal performance impact. These findings suggest that ChunkKV represents a significant advancement in KV cache compression technology, offering an effective solution for deploying LLMs in resource-constrained environments while maintaining high-quality outputs and paving the way for future developments in efficient, semantically aware LLM inference.

ETHICS STATEMENT

Our study does not involve human subjects, data collection from individuals, or experiments on protected groups. The models and datasets used in this work are publicly available and widely used in the research community. We have made efforts to ensure our experimental design and reporting of results are fair, unbiased, and do not misrepresent the capabilities or limitations of the methods presented.

In our work on KV cache compression for large language models, we acknowledge the potential broader impacts of improving efficiency in AI systems. While our method aims to reduce computational resources and potentially increase accessibility of these models, we recognize that more efficient language models could also lead to increased deployment and usage, which may have both positive and negative societal implications. We encourage further research and discussion on the responsible development and application of such technologies.

We declare no conflicts of interest that could inappropriately influence our work. All experiments were conducted using publicly available resources, and our code will be made available to ensure reproducibility. We have made every effort to cite relevant prior work appropriately and to accurately represent our contributions in the context of existing research.

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

## A  IN-DEPTH ANALYSIS OF CHUNKKV VS. DISCRETE TOKEN METHODS

### A.1  QUANTITATIVE ANALYSIS

To rigorously evaluate the effectiveness of ChunkKV compared to discrete token-based methods, we conducted systematic experiments using a LLaMA-3-8B-Instruct model. We randomly selected 100 sequences from the each sub-category of LongBench dataset and analyzed two key metrics across different model layers: KV cache L1 loss and attention cosine similarity. For each sequence, we: 1. Computed the full KV cache and attention patterns without compression as ground truth. 2. Applied ChunkKV, SnapKV, and H2O compression methods with a fixed 10% compression ratio, and the parameters of the three methods are set the same as in Table 4. 3. Measured the differences between compressed and uncompressed versions.

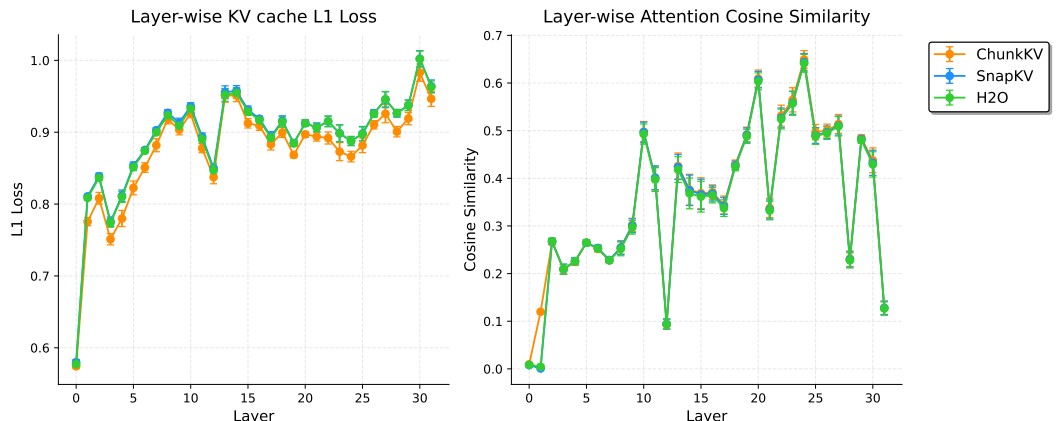

Figure 7: Layer-wise comparison of L1 loss and attention cosine similarity between ChunkKV and discrete token-based methods in Single-Document QA sub-category of LongBench.

**Results Analysis**    As shown in Figure 7, ChunkKV demonstrates superior performance across both metrics:

- **KV Cache L1 Loss:** ChunkKV achieves consistently lower L1 loss compared to SnapKV and H2O, particularly in the early and middle layers (layers 5-25). This indicates better preservation of the original KV cache information through the semantic chunk-based approach.

- **Attention Cosine Similarity:** ChunkKV exhibits higher similarity scores across most layers, with notably strong performance in layers 0-5 and 20-30. This suggests better preservation of attention relationships between tokens, which is crucial for maintaining semantic understanding.

To quantify these improvements, we calculated average metrics across all layers, as shown in Table 8. ChunkKV achieves both the lowest L1 loss and highest attention cosine similarity, outperforming both baseline methods.

**Significance of Results**    While the improvements may appear modest in absolute terms (approximately 2% in L1 loss and 1.5% in cosine similarity), their practical significance is substantial. These metrics reflect the model's ability to maintain crucial semantic relationships and attention patterns, which are essential for complex reasoning tasks. The consistent improvements across different sequences demonstrate that preserving semantic chunks leads to better information retention than selecting individual tokens.

Table 8: Detailed comparison of KV cache metrics across different task categories in LongBench.

| Method | Single-Document QA | Multi-Document QA | Summarization | Few-shot Learning | Synthetic & Code |
|---|---|---|---|---|---|
| KV Cache L1 Loss ↓ | | | | | |
| **ChunkKV** | **0.8741** | **0.8748** | **0.8770** | **0.8861** | **0.8726** |
| SnapKV | 0.8921 | 0.8933 | 0.8930 | 0.8917 | 0.8938 |
| H2O | 0.8905 | 0.8917 | 0.8913 | 0.8906 | 0.8915 |
| Attention Score Cosine Similarity ↑ | | | | | |
| **ChunkKV** | **0.3567** | **0.3651** | **0.3841** | **0.4330** | **0.3805** |
| SnapKV | 0.3513 | 0.3594 | 0.3771 | 0.4305 | 0.3759 |
| H2O | 0.3491 | 0.3572 | 0.3750 | 0.4284 | 0.3740 |

The enhanced performance is particularly evident in the middle layers of the model, which are typically responsible for higher-level semantic processing. This provides concrete evidence for why ChunkKV achieves superior performance on downstream tasks compared to discrete token-based methods.

## A.2 HYPOTHETICAL SCENARIO

To provide a deeper understanding of ChunkKV's effectiveness compared to discrete token-based methods, we present a detailed analysis using a hypothetical scenario. This analysis aims to illustrate the fundamental differences between these approaches and explain why ChunkKV is more effective at preserving semantic information in long contexts.

Consider a comprehensive document that contains detailed information on various animals, including their habitats, diets, and behaviors. A user asks the question "What do pandas eat in the wild?"

Both ChunkKV and discrete token-based methods would use this question to calculate observation scores for the document. However, their approaches to selecting and retaining information differ significantly.

### A.2.1 DISCRETE TOKEN-BASED METHOD

A discrete token-based method might identify and retain individual tokens with high relevance scores, such as:

- "pandas","eat", "bamboo", "wild", "diet", "food"

Although these tokens are relevant, they lack context and coherence. The method might discard other essential tokens that provide crucial context or complete the information.

### A.2.2 CHUNKKV METHOD

In contrast, ChunkKV would identify and retain semantically meaningful chunks, such as:

- "In the wild, pandas primarily eat bamboo shoots and leaves"
- "Their diet consists of 99% bamboo, but they occasionally consume other vegetation"
- "Wild pandas may also eat small rodents or birds when available"

By preserving these chunks, ChunkKV maintains not only the relevant keywords but also their contextual relationships and additional pertinent information.

## A.3 COMPARATIVE ANALYSIS

The advantages of ChunkKV become evident when we consider how these retained pieces of information would be used in subsequent processing:

1. **Contextual Understanding**: Discrete tokens require the model to reconstruct meaning from isolated words, which could lead to ambiguity. ChunkKV provides complete phrases or sentences, allowing for immediate and accurate comprehension.

2. **Semantic Coherence**: ChunkKV preserves the semantic relationships within a chunk, crucial to understanding nuances such as the difference between primary and occasional food sources for pandas.

3. **Information Density**: A single chunk can contain multiple relevant tokens in their proper context, potentially retaining more useful information within the same compressed cache size compared to discrete methods.

4. **Reduced Ambiguity**: Discrete methods might retain the token "eat" from various sentences about different animals. ChunkKV ensures that "eat" is preserved specifically in the context of pandas in the wild.

5. **Temporal and Logical Flow**: ChunkKV can maintain the sequence of ideas present in the original text, preserving any temporal or logical progression that may be crucial for understanding.

## A.4 IMPLICATIONS FOR MODEL PERFORMANCE

This analysis suggests several key implications for model performance:

- **Improved Accuracy**: By retaining contextually rich information, ChunkKV enables more accurate responses to queries, especially those requiring nuanced understanding.

- **Enhanced Long-context Processing**: Preservation of semantic chunks allows for better handling of long-range dependencies and complex reasoning tasks.

- **Reduced Computational Overhead**: Although both methods compress the KV cache, ChunkKV's approach may reduce the need for extensive context reconstruction, potentially improving inference efficiency.

- **Versatility**: The chunk-based approach is likely to be more effective across a wide range of tasks and domains as it preserves the natural structure of language.

This in-depth analysis demonstrates why ChunkKV is more effective in preserving semantic information in long contexts. By retaining coherent chunks of text, it provides language models with more contextually rich and semantically complete information, leading to improved performance in tasks that require deep understanding and accurate information retrieval from extensive documents.

## B ADDITIONAL EXPERIMENTS

### B.1 EFFICIENCY

We evaluated the latency and throughput of ChunkKV compared to FullKV using LLaMA3-8B-Instruct on an A40 GPU. All experiments were conducted with a batch size of 1 and inference was performed using Flash Attention 2, each experiment was repeated 10 times and the average latency and throughput were reported. The results demonstrate that ChunkKV not only maintains competitive performance but also achieves improved efficiency, which is further enhanced by layer-wise index reuse.

The results in Table 9 highlight several key findings:

- ChunkKV consistently outperforms FullKV across all configurations, achieving latency improvements ranging from 6.2% to 18.6%.

- The layer-wise index reuse strategy (ChunkKV_reuse) further boosts performance, achieving up to a 20.7% reduction in latency.

- Throughput improvements are particularly notable for longer input sequences, with ChunkKV_reuse delivering up to a 26.5% improvement over FullKV.

Table 9: Latency and throughput comparison between ChunkKV and FullKV under different input-output configurations. Percentages in parentheses indicate improvements over FullKV baseline.

| Method | Sequence Length | | Performance Metrics | |
|---|---|---|---|---|
| | Input | Output | Latency(s) ↓ | Throughput(T/S) ↑ |
| FullKV | 4096 | 1024 | 43.60 | 105.92 |
| ChunkKV | 4096 | 1024 | 37.52 (13.9%) | 118.85 (12.2%) |
| ChunkKV_reuse | 4096 | 1024 | **37.35** (14.3%) | **124.09** (17.2%) |
| FullKV | 4096 | 4096 | 175.50 | 37.73 |
| ChunkKV | 4096 | 4096 | 164.55 (6.2%) | 40.58 (7.6%) |
| ChunkKV_reuse | 4096 | 4096 | **162.85** (7.2%) | **41.12** (9.0%) |
| FullKV | 8192 | 1024 | 46.48 | 184.08 |
| ChunkKV | 8192 | 1024 | 37.83 (18.6%) | 228.96 (24.4%) |
| ChunkKV_reuse | 8192 | 1024 | **36.85** (20.7%) | **232.99** (26.5%) |
| FullKV | 8192 | 4096 | 183.42 | 55.93 |
| ChunkKV | 8192 | 4096 | 164.78 (10.2%) | 65.14 (16.5%) |
| ChunkKV_reuse | 8192 | 4096 | **162.15** (11.6%) | **66.05** (18.1%) |

These efficiency gains are even more pronounced with longer input sequences, demonstrating that ChunkKV is particularly well-suited for processing long-context inputs while maintaining minimal memory overhead.

## B.2 LAYER-WISE INDEX REUSE

### B.2.1 LAYER-WISE INDEX SIMILARITY

This section details the experiment of layer-wise index reuse similarity described in Section 3.2.2. The inference prompt is randomly selected from the LongBench benchmark, and the preserved indices for H2O, SnapKV, and ChunkKV are saved in the log file. For multi-head attention, only the indices of the first head are saved. PyramidKV, which has varying preserved index sizes across different layers, is not applicable for this experiment. Then we calculate the Jaccard similarity of the preserved indices of adjacent layers for different models. Table 10 shows the Jaccard similarity of the preserved indices of adjacent layers for different models.

Table 10: Retained KV Cache Indices Similarity of Adjacent Layers for Different Models.

| Method | H2O | SnapKV | ChunkKV |
|---|---|---|---|
| LLaMA-3-8B-Instruct | 25.31% | 27.95% | **57.74%** |
| Qwen2-7B-Instruct | 14.91% | 16.50% | **44.26%** |
| Mistral-7B-Instruct | 15.15% | 15.78% | **52.16%** |

Figures 8-10 (LLaMA-3-8B-Instruct), 11-13 (Mistral-7B-Instruct), and 14-16 (Qwen2-7B-Instruct) display the heatmaps of layer-wise indices similarity of the preserved KV cache indices by H2O, SnapKV and ChunkKV on different models. The pattern of the layer-wise indices similarity heatmap is consistent across different models, aligning with our findings in Section 3.2.2.

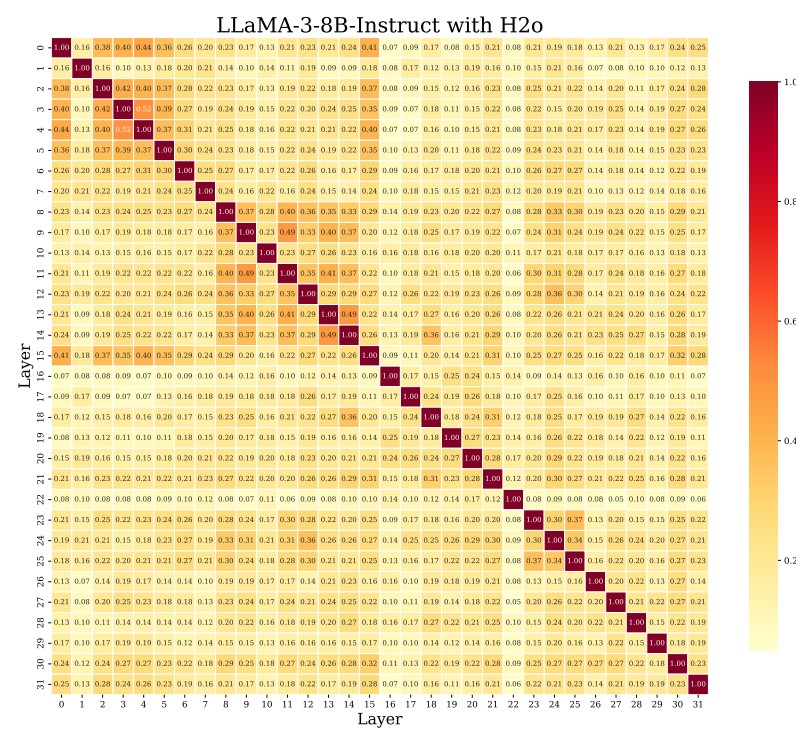

Figure 8: Layer-wise similarity heatmaps of the preserved KV cache indices by H2O on LLaMA-3-8B-Instruct

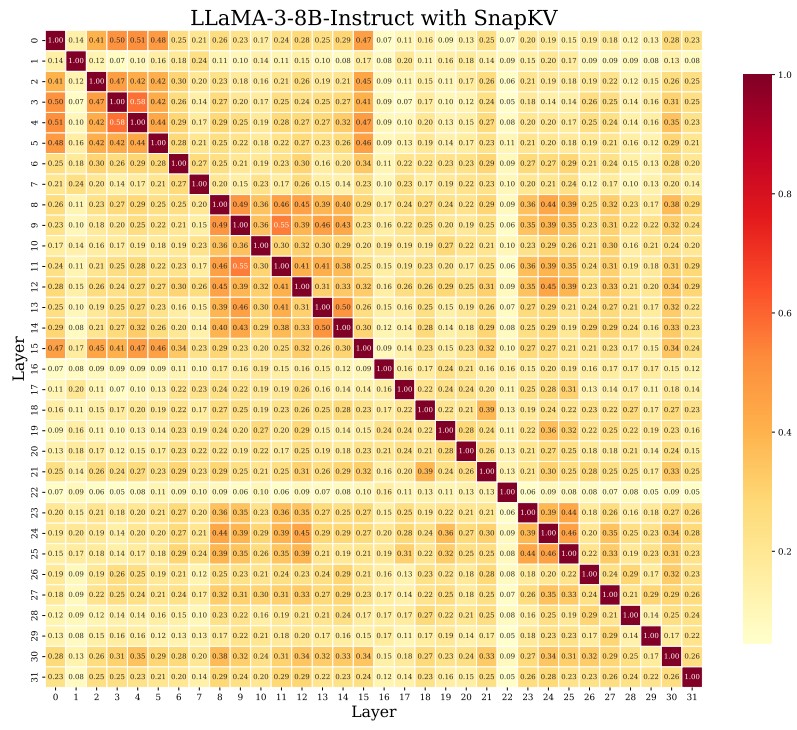

Figure 9: Layer-wise similarity heatmaps of the preserved KV cache indices by SnapKV on LLaMA-3-8B-Instruct

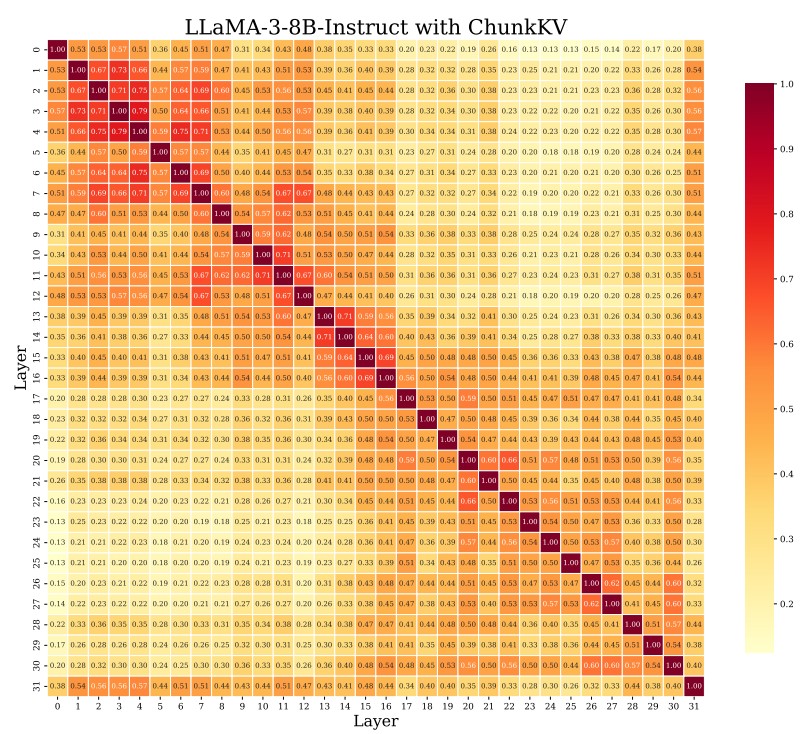

Figure 10: Layer-wise similarity heatmaps of the preserved KV cache indices by ChunkKV on LLaMA-3-8B-Instruct

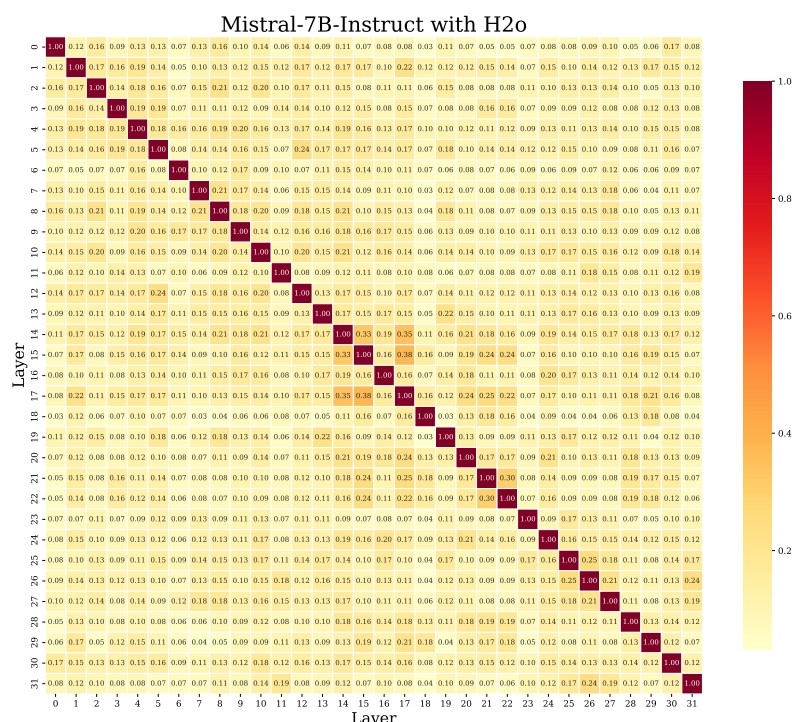

Figure 11: Layer-wise similarity heatmaps of the preserved KV cache indices by H2O on Mistral-7B-Instruct

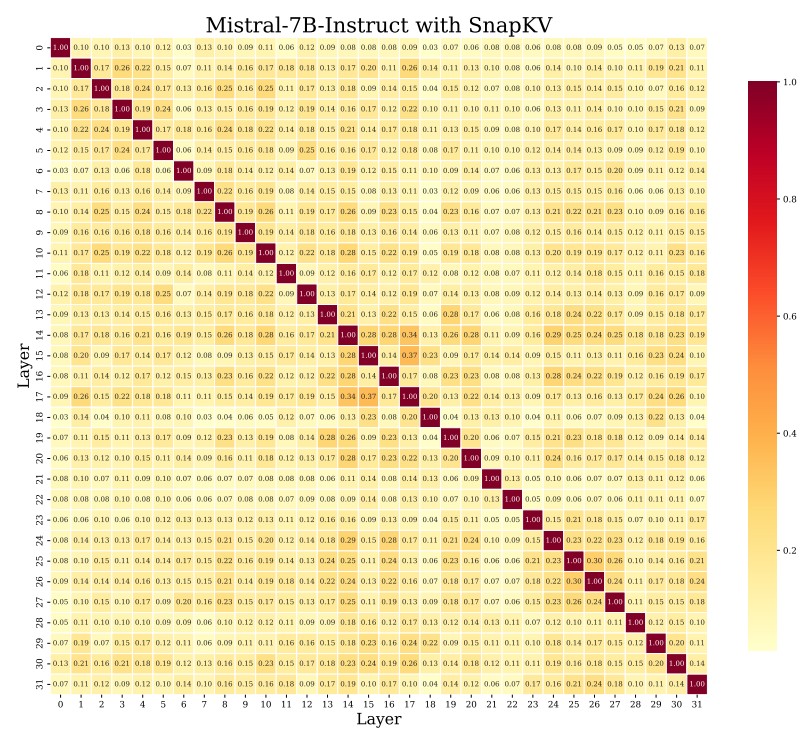

Figure 12: Layer-wise similarity heatmaps of the preserved KV cache indices by SnapKV on Mistral-7B-Instruct

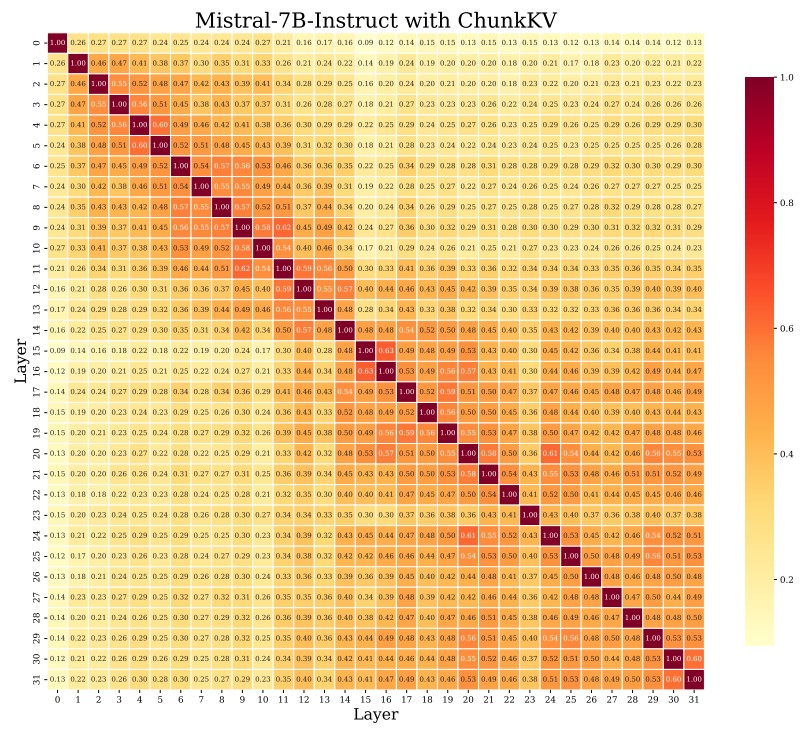

Figure 13: Layer-wise similarity heatmaps of the preserved KV cache indices by ChunkKV on Mistral-7B-Instruct

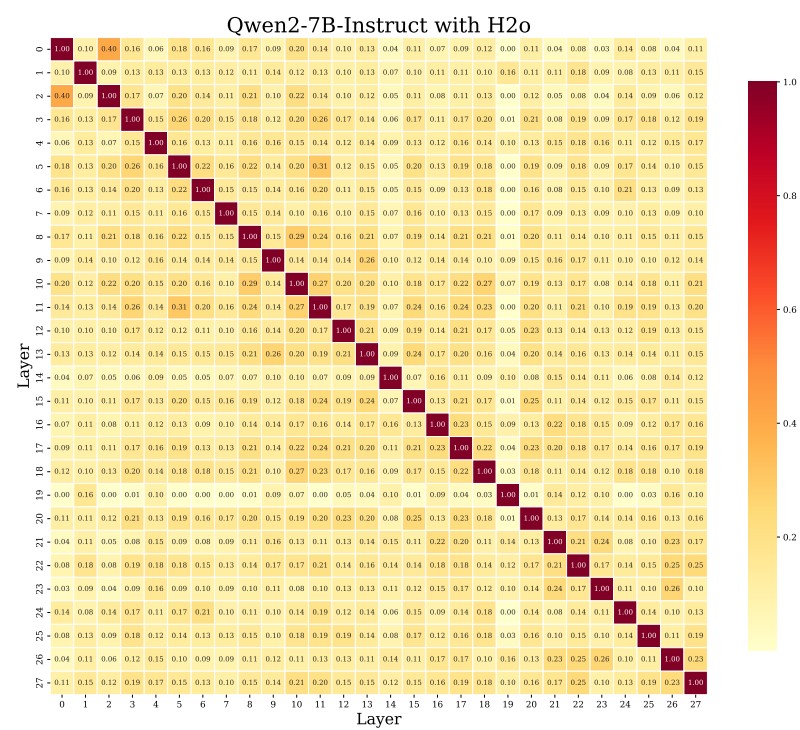

Figure 14: Layer-wise similarity heatmaps of the preserved KV cache indices by H2O on Qwen2-7B-Instruct

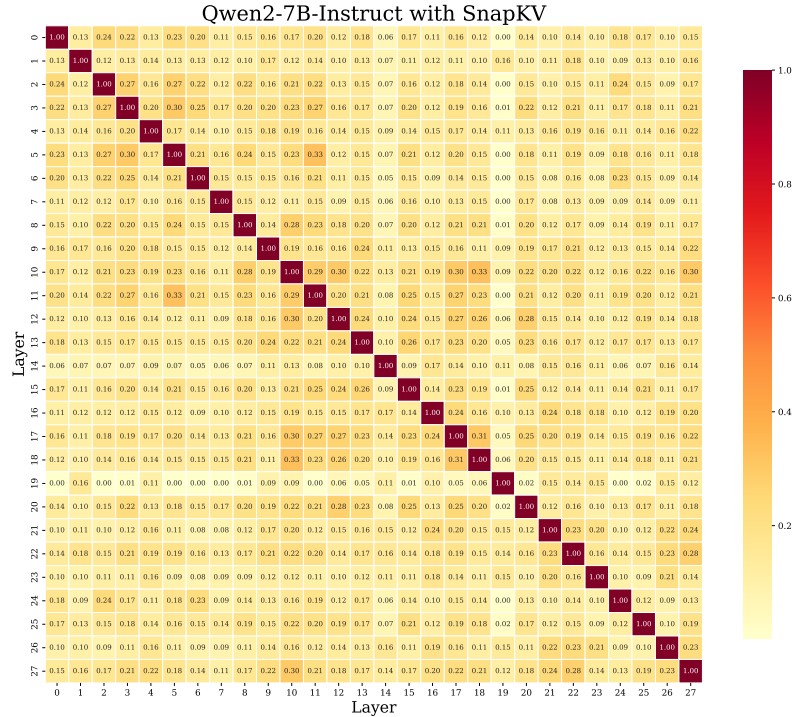

Figure 15: Layer-wise similarity heatmaps of the preserved KV cache indices by SnapKV on Qwen2-7B-Instruct

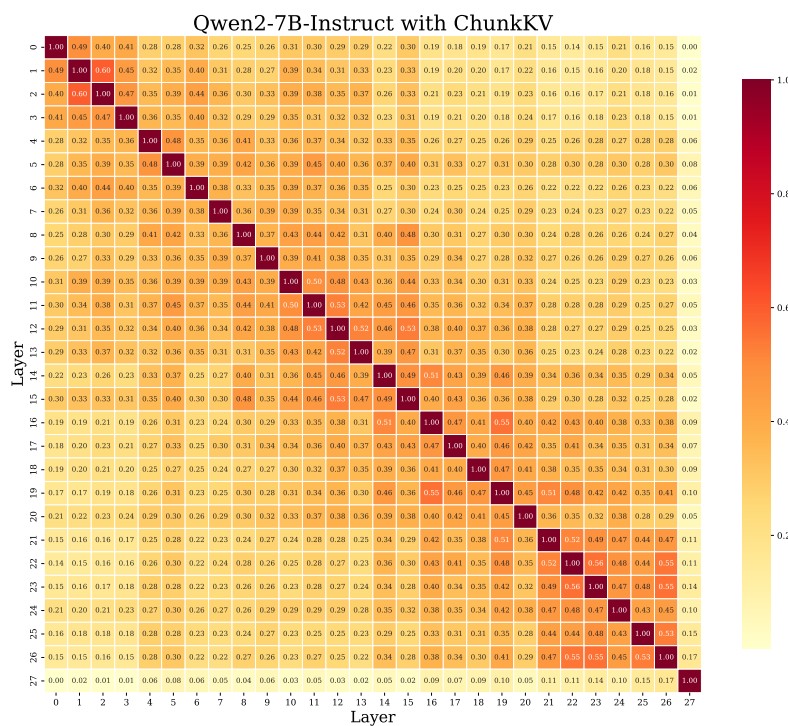

Figure 16: Layer-wise similarity heatmaps of the preserved KV cache indices
by ChunkKV on Qwen2-7B-Instruct

### B.2.2 INDEX REUSE PERFORMANCE

Figure 17 illustrates the performance of ChunkKV with varying index reuse layers on the GSM8K benchmark. The experiment reveals that math problems are more sensitive to index reuse layers compared to LongBench. Both LLaMA3-8B-Instruct and Qwen2-7B-Instruct exhibit significant performance degradation, with LLaMA3-8B-Instruct experiencing a steeper decline after two layers of index reuse than Qwen2-7B-Instruct. This suggests that the Qwen2-7B-Instruct model may be more robust to index reuse.

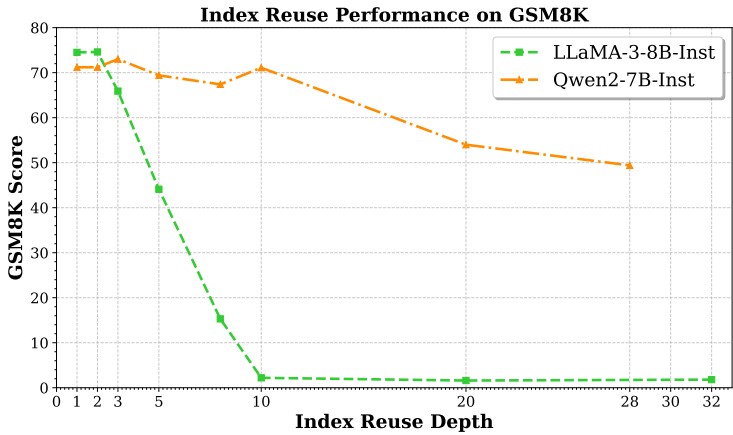

Figure 17: GSM8K Performance Comparison with different index reuse layers

Table 11 shows the performance of ChunkKV with different numbers of index reuse layers in GSM8K. The number of index reuse layers is set from 1 to the number of layers in the model, where a index reuse layer of 1 corresponds to the normal ChunkKV without index reuse, and 28/32

is the maximum number of layers for LLaMA-3-8B-Instruct and Qwen2-7B-Instruct. ==The significant performance drop of LLaMA-3-8B-Instruct raises another question: whether the KV cache compression method is more sensitive to the model's mathematical reasoning ability.==

Table 11: Reusing Indexing Performance Comparison on GSM8K

| Model | Number of Index Reuse Layers | | | | | | | |
|---|---|---|---|---|---|---|---|---|
| | 1 | 2 | 3 | 5 | 8 | 10 | 20 | 28/32 |
| LLaMA-3-8B-Instruct | 74.5 | 74.6 | 65.9 | 44.1 | 15.3 | 2.20 | 1.60 | 1.80 |
| Qwen2-7B-Instruct | 71.2 | 71.2 | 73.0 | 69.4 | 67.4 | 71.1 | 54.0 | 49.4 |

## B.3 LONGBENCH

The Table 12 shows the average performance of KV cache compression methods in the LongBench English subtask categories. The ChunkKV achieves the best performance on the overall average, and the Multi-Document QA category, which supports that chunk method is more effective for semantic preservation.

Table 12: Comprehensive performance comparison of KV cache compression methods across Long-Bench English subtasks. Results are shown for various models and tasks, highlighting the effectiveness of different compression techniques.

| Method | Single-Document QA | | | Multi-Document QA | | | Summarization | | | Few-shot Learning | | | Synthetic | | Code | | Avg. ↑ |
|---|---|---|---|---|---|---|---|---|---|---|---|---|---|---|---|---|---|
| | NrtvQA | Qasper | MF-en | HotpotQA | 2WikiMQA | Musique | GovReport | QMSum | MultiNews | TREC | TriviaQA | SAMSum | PCount | PRe | Lcc | RB-P | |
| Avg len | 18,409 | 3,619 | 4,559 | 9,151 | 4,887 | 11,214 | 8,734 | 10,614 | 2,113 | 5,177 | 8,209 | 6,258 | 11,141 | 9,289 | 1,235 | 4,206 | |
| LlaMa-3-8B-Instruct, KV Size = Full | | | | | | | | | | | | | | | | | |
| FullKV | 25.70 | 29.75 | 41.12 | 45.55 | 35.87 | 22.35 | 25.63 | 23.03 | 26.21 | 73.00 | 90.56 | 41.88 | 4.67 | 69.25 | 58.05 | 50.77 | 41.46 |
| LlaMa-3-8B-Instruct, KV Size Compression Ratio = 10% | | | | | | | | | | | | | | | | | |
| StreamingLLM | 20.62 | 13.09 | 22.10 | 36.31 | 28.01 | 15.61 | 21.47 | 21.05 | 19.39 | 62.00 | 84.18 | 40.27 | 4.62 | 69.10 | 58.84 | 55.26 | 35.74 |
| H2O | 24.80 | 17.32 | 31.80 | 40.84 | 33.28 | 18.90 | 22.29 | 22.29 | 21.82 | 40.00 | 90.51 | 40.55 | 5.79 | **69.50** | 58.04 | 55.26 | 37.06 |
| SnapKV | 25.08 | 22.02 | **37.95** | 43.36 | 35.08 | 20.29 | 22.94 | 22.64 | 21.37 | 71.00 | 90.47 | 40.15 | 5.66 | 69.25 | 58.69 | 56.50 | 40.15 |
| PyramidKV | **25.58** | 20.77 | 35.85 | **43.80** | 33.03 | **21.45** | **23.68** | 22.26 | **21.85** | 71.50 | 90.47 | **41.66** | 5.84 | 69.25 | 58.52 | 55.91 | 40.08 |
| **ChunkKV** | 24.89 | **22.96** | 37.64 | 43.27 | **36.45** | 20.65 | 22.80 | **22.97** | 20.82 | 71.50 | **90.52** | 40.83 | **5.93** | 69.00 | **60.49** | **57.48** | **40.51** |
| LlaMa-3-8B-Instruct, KV Size Compression Ratio = 20% | | | | | | | | | | | | | | | | | |
| StreamingLLM | 23.35 | 18.97 | 32.94 | 42.39 | 29.37 | 18.76 | **25.78** | 21.92 | **25.16** | 71.00 | 88.85 | 40.82 | 5.04 | 69.00 | 56.46 | 51.12 | 38.80 |
| H2O | 25.60 | 21.88 | 35.36 | 42.06 | 32.68 | 19.72 | 23.54 | 22.77 | 22.72 | 45.50 | **90.57** | **41.67** | **5.51** | 69.25 | 54.97 | 50.95 | 37.79 |
| SnapKV | 25.50 | 25.95 | 38.43 | 44.12 | 35.38 | 20.49 | 24.85 | 23.36 | 23.51 | **72.50** | 90.52 | 40.91 | 5.23 | 69.25 | 56.74 | 51.75 | 40.53 |
| PyramidKV | 25.36 | 26.88 | 37.99 | 44.21 | **35.65** | **21.43** | 25.52 | **23.43** | 23.47 | 72.00 | 90.56 | 41.45 | 5.26 | **69.50** | 56.55 | 50.93 | 40.63 |
| **ChunkKV** | **26.13** | **28.43** | **38.59** | **44.46** | 34.13 | 21.06 | 24.72 | 23.11 | 22.91 | 71.50 | 90.56 | 41.51 | 5.09 | 69.00 | **58.17** | **52.51** | **40.74** |
| LlaMa-3-8B-Instruct, KV Size Compression Ratio = 30% | | | | | | | | | | | | | | | | | |
| StreamingLLM | 24.49 | 22.53 | 35.30 | **44.33** | 32.81 | 19.00 | **27.12** | 22.19 | **25.93** | 72.50 | 89.84 | 41.75 | **5.41** | 69.00 | 60.40 | 55.13 | 40.48 |
| H2O | 25.87 | 23.03 | 37.06 | 43.71 | 33.68 | 20.93 | 24.56 | 23.14 | 23.58 | 50.50 | **90.77** | 41.96 | 4.91 | 69.25 | 59.38 | 55.39 | 39.23 |
| SnapKV | 25.15 | 28.75 | **39.28** | 43.57 | 36.16 | 21.58 | 25.56 | **23.19** | 24.30 | **73.00** | 90.52 | 41.70 | 4.96 | 69.25 | 60.27 | 55.74 | 41.43 |
| PyramidKV | 25.42 | 27.91 | 38.81 | 44.15 | **36.28** | **21.72** | 26.23 | 23.10 | 24.15 | 72.00 | 90.56 | 41.87 | 4.67 | **69.50** | 60.09 | 55.19 | 41.37 |
| **ChunkKV** | **25.88** | **29.58** | 38.99 | 43.94 | 34.16 | 21.70 | 26.50 | 23.15 | 23.95 | 72.00 | 90.56 | **42.47** | 5.34 | 69.25 | **61.68** | **56.35** | **41.59** |
| Mistral-7B-Instruct-v0.3, KV Size = Full | | | | | | | | | | | | | | | | | |
| FullKV | 29.07 | 41.58 | 52.88 | 49.37 | 39.01 | 28.58 | 34.93 | 25.68 | 27.74 | 76.00 | 88.59 | 47.59 | 6.00 | 98.50 | 61.41 | 62.39 | 48.08 |
| Mistral-7B-Instruct-v0.3, KV Size Compression Ratio = 10% | | | | | | | | | | | | | | | | | |
| StreamingLLM | 25.15 | 25.47 | 30.08 | 44.39 | 32.49 | 19.40 | 24.11 | 20.85 | 20.85 | 65.00 | 88.21 | 44.83 | 4.50 | 79.50 | 59.48 | 58.82 | 40.11 |
| H2O | 29.35 | 33.39 | 50.39 | 49.58 | 36.76 | 27.42 | 25.16 | 24.75 | 22.12 | 42.00 | 89.00 | **47.04** | 5.50 | 98.50 | 57.58 | 59.24 | 43.61 |
| SnapKV | 28.54 | **36.88** | 53.42 | 50.15 | 38.17 | 27.99 | 26.67 | **25.21** | **22.33** | 72.00 | 89.36 | 45.44 | **5.50** | **99.00** | 59.79 | **61.63** | 46.38 |
| PyramidKV | 29.40 | 35.39 | 52.96 | 49.93 | 38.67 | 28.63 | **27.59** | 24.99 | 22.77 | 74.00 | **90.02** | 46.07 | 4.00 | 98.50 | 58.54 | 60.88 | 46.39 |
| **ChunkKV** | **29.75** | 36.82 | **53.99** | **50.33** | **38.72** | **29.01** | 27.03 | 24.76 | 21.42 | **76.00** | 88.73 | 46.49 | 5.00 | 98.00 | **59.98** | 61.47 | **46.71** |
| Qwen2-7B-Instruct, KV Size = Full | | | | | | | | | | | | | | | | | |
| FullKV | 25.11 | 42.64 | 44.29 | 14.25 | 13.22 | 9.08 | 36.38 | 23.43 | 26.53 | 77.00 | 89.99 | 44.88 | 6.75 | 75.92 | 60.17 | 61.84 | 40.71 |
| Qwen2-7B-Instruct, KV Size Compression Ratio = 10% | | | | | | | | | | | | | | | | | |
| StreamingLLM | 25.15 | 45.42 | 41.46 | 13.66 | 11.95 | 8.72 | 32.79 | 21.49 | **26.24** | 77.50 | 89.15 | 44.54 | 7.50 | 50.50 | 60.03 | 60.91 | 38.56 |
| H2O | 26.17 | 44.33 | 42.54 | 12.81 | 12.46 | 9.15 | **33.24** | 22.69 | 25.94 | 76.50 | 89.44 | 44.32 | 8.00 | 76.00 | **61.28** | **62.39** | 40.45 |
| SnapKV | 26.84 | **45.96** | **45.79** | 14.27 | **13.35** | 9.91 | 32.62 | 22.70 | 25.83 | 77.00 | 89.19 | 44.71 | 7.50 | 71.50 | 60.35 | 61.37 | 40.55 |
| PyramidKV | **27.51** | 44.45 | 43.59 | 13.35 | 13.13 | 9.12 | 32.28 | 22.60 | 25.45 | 77.00 | **89.44** | 44.53 | 7.00 | 73.50 | 60.91 | 61.24 | 40.31 |
| **ChunkKV** | 26.48 | 44.19 | 45.04 | **15.94** | 12.60 | **10.52** | 32.38 | **22.87** | 25.91 | **77.50** | 89.22 | **44.78** | **8.50** | **76.50** | 60.64 | 61.32 | **40.88** |

## B.4 NEEDLE-IN-A-HAYSTACK

Figure 18 and 19 visualizes the performance of ChunkKV on the NIAH benchmark for LLaMA-3-8B-Instruct and Mistral-7B-Instruct with a KV cache size of 128 under 8k and 32k context length. The performance of ChunkKV is consistently better as the context length increases.

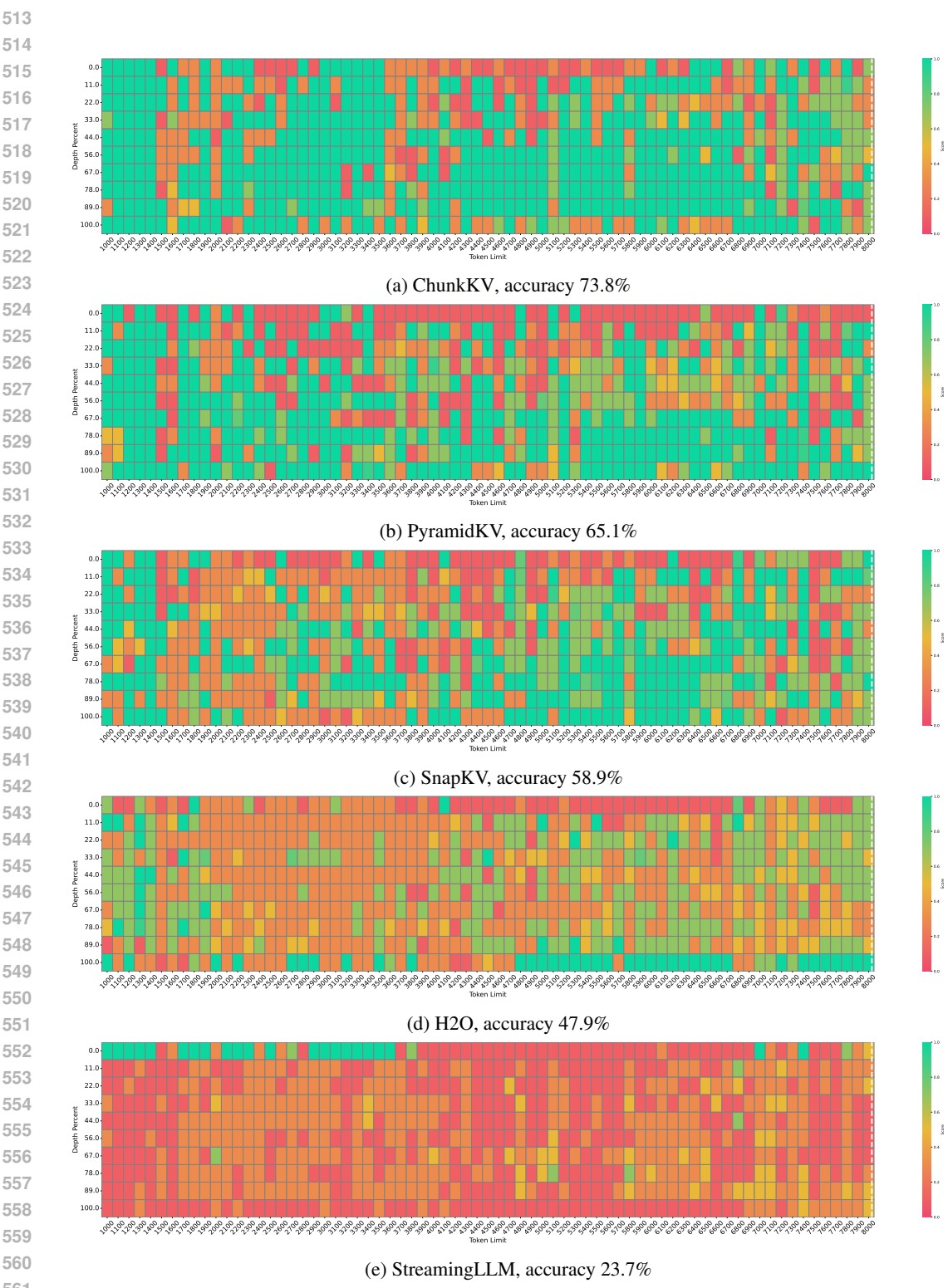

Figure 18: NIAH benchmark for LLaMA-3-8B-Instruct with KV cache size=128 under 8k context length

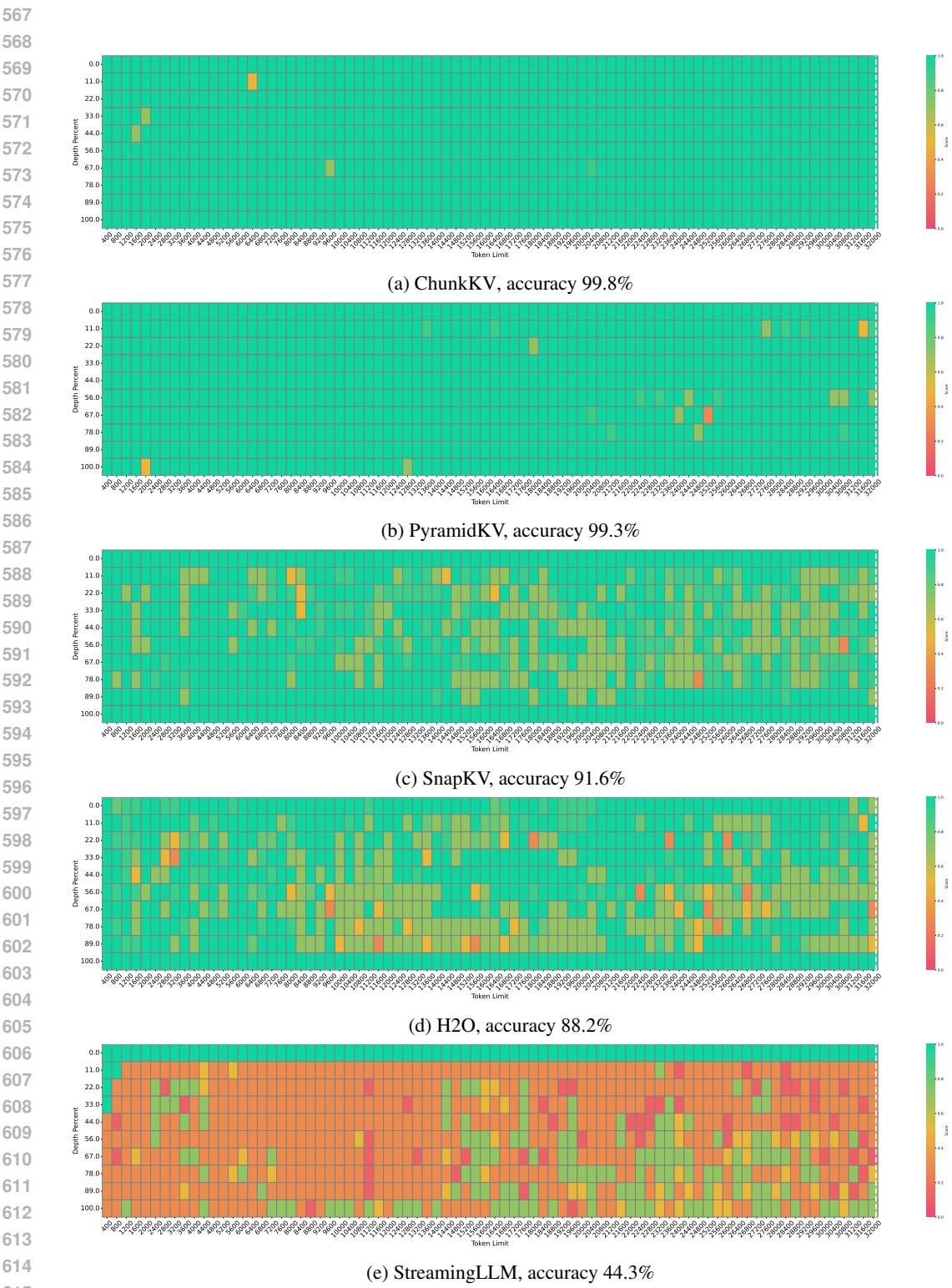

Figure 19: NIAH benchmark for Mistral-7B-Instruct with KV cache size=128 under 32k context length

Table 13 shows the performance of ChunkKV in the NIAH data set with different KV cache sizes on LLaMA-3-8B-Instruct.

Table 13: NIAH Performance Comparison with Different KV Cache Sizes

| Method | Size = 96 | Size = 128 | Size = 256 | Size = 512 |
|--------|-----------|------------|------------|------------|
| StreamingLLM | 21.5 | 23.7 | 28.0 | 32.0 |
| H2O | 41.0 | 47.9 | 61.7 | 68.6 |
| SnapKV | 56.2 | 58.9 | 68.8 | 71.2 |
| PyramidKV | 63.2 | 65.1 | 69.5 | 72.6 |
| **ChunkKV** | **70.3** | **73.8** | **74.1** | **74.5** |
| FullKV | 74.6 | 74.6 | 74.6 | 74.6 |

### B.5 CHUNK SIZE

Table 14 and 15 show the performance of ChunkKV with different comperession ratios and different chunk sizes on the LongBench and NIAH. We conducted extensive experiments across different compression ratios and KV cache sizes to shows the effectiveness of ChunkKV and the chunk size is robust.

Table 14: LongBench Performance with Different Chunk Sizes and Compression Ratios for LLaMA-3-8B-Instruct

| Compression | Chunk Size | | | | | | |
|-------------|------|------|------|------|------|------|------|
| Rate | 1 | 3 | 5 | 10 | 15 | 20 | 30 |
| 10% | 37.32 | 40.49 | 40.47 | **40.51** | 40.21 | 40.05 | 39.57 |
| 20% | 38.80 | 40.66 | 40.57 | **40.74** | 40.53 | 40.46 | 40.04 |
| 30% | 39.23 | 41.02 | 41.29 | **41.59** | 41.38 | 41.33 | 41.02 |

Table 15: NIAH Performance with Different Chunk Sizes and KV Cache Sizes for LLaMA-3-8B-Instruct

| KV Cache | Chunk Size | | | | | | |
|----------|------|------|------|------|------|------|------|
| Size | 1 | 3 | 5 | 10 | 15 | 20 | 30 |
| 96 | 41.0 | 63.2 | 65.2 | **70.3** | 67.2 | 65.3 | 53.1 |
| 128 | 47.9 | 65.6 | 69.1 | **73.8** | 72.3 | 72.0 | 71.2 |
| 256 | 61.7 | 70.3 | 71.2 | **74.1** | 73.2 | 72.3 | 71.1 |
| 512 | 68.6 | 72.6 | 72.5 | **74.5** | 74.3 | 74.0 | 72.6 |

Table 16 shows the performance of ChunkKV with different chunk size on the LongBench benchmark.

Table 17 shows the performance of ChunkKV with different chunk size on the GSM8K benchmark. Figure 20 shows that the ChunkKV with different chunk sizes on GSM8K displays the same curve pattern as LongBench. The CoT prompt length for GSM8K is only 1K tokens, so the optimal chunk size range is smaller.

Table 16: LongBench Performance Comparison with different chunk sizes

| Model | Chunk Size | | | | | Full KV |
|---|---|---|---|---|---|---|
| | 3 | 5 | 10 | 20 | 30 | |
| LLaMA-3-8B-Instruct | 40.49 | 40.47 | **40.51** | 40.05 | 39.57 | 41.46 |
| Mistral-7B-Instruct | 46.45 | 46.51 | **46.71** | 46.42 | 45.98 | 48.08 |
| Qwen2-7B-Instruct | 40.38 | 40.33 | 40.66 | **40.88** | 40.73 | 40.71 |

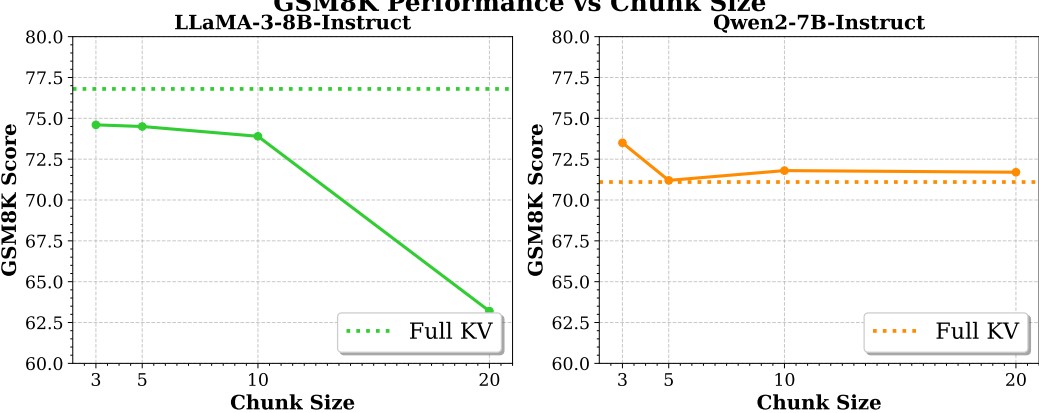

Figure 20: GSM8K Performance Comparison
with different chunk size

Table 17: GSM8K Performance Comparison with different chunk sizes

| Model | Chunk Size | | | | Full KV |
|---|---|---|---|---|---|
| | 3 | 5 | 10 | 20 | |
| LLaMA-3-8B-Instruct | **74.6** | 74.5 | 73.9 | 63.2 | 76.8 |
| Qwen2-7B-Instruct | **73.5** | 71.2 | 71.8 | 71.7 | 71.1 |

### B.6 MULTI-LINGUAL

Table 18 is the Chinese support model Qwen2-7B-Instruct evaluated on the LongBench Chinese subtask, where ChunkKV achieves better performance than other compression methods and the full KV cache performance. Both the English and Chinese results indicate that ChunkKV is a promising approach for maintaining crucial information in the KV cache.

## C ADDITIONAL RELATED WORK

**KV cache sharing** Recent work has explored various strategies for sharing KV caches across transformer layers. Layer-Condensed KV Cache (LCKV) (Wu & Tu, 2024) computes KVs only for the top layer and pairs them with queries from all layers, while optionally retaining standard attention for a few top and bottom layers to mitigate performance degradation. Similarly, You Only Cache Once (YOCO) (Sun et al., 2024) computes KVs exclusively for the top layer but pairs them with queries from only the top half of layers, employing efficient attention in the bottom layers to maintain a constant cache size. In contrast, Cross-Layer Attention (CLA) (Brandon et al., 2024) divides layers into groups, pairing queries from all layers in each group with KVs from that group's bottom layer. MiniCache (Liu et al., 2024a) introduces a novel method that merges layer-wise KV caches while enabling recovery during compute-in-place operations, optimizing KV cache size. These methods

Table 18: Performance comparison of Chinese subtask on LongBench for Qwen2-7B-Instruct.

| Method | Single-Document QA | Multi-Document QA | Summarization | Few-shot Learning | Synthetic | |
|---|---|---|---|---|---|---|
| | MF-zh | DuReader | VCSum | LSHT | PR-zh | Avg. ↑ |
| Avg len | 6,701 | 15,768 | 15,380 | 22,337 | 6,745 | |
| Qwen2-7B-Instruct, KV Size = Full | | | | | | |
| FullKV | 39.17 | 23.63 | 16.21 | 43.50 | 70.50 | 38.60 |
| Qwen2-7B-Instruct, KV Size Compression Ratio = 10% | | | | | | |
| StreamingLLM | 38.05 | **23.24** | 15.92 | 40.50 | 44.50 | 32.44 |
| H2O | 37.99 | 19.58 | 16.16 | 41.67 | 67.35 | 36.55 |
| SnapKV | 44.25 | 20.27 | 16.24 | **44.50** | 68.10 | 38.67 |
| PyramidKV | 36.57 | 20.56 | 16.15 | 43.50 | 66.50 | 36.55 |
| **ChunkKV** | **45.92** | 20.15 | **16.37** | 43.75 | **71.10** | **39.45** |

illustrate various trade-offs between computation, memory usage, and model performance when sharing KV caches across transformer layers.

**Long-Context Benchmarks**   The landscape of long-context model benchmarks has evolved to encompass a wide range of tasks, with particular emphasis on retrieval and comprehension capabilities. Benchmarks for understanding have made significant strides, with ∞-Bench (Zhang et al., 2024a) pushing the boundaries by presenting challenges that involve more than 100,000 tokens. LongBench (Bai et al., 2024) has introduced bilingual evaluations, addressing tasks such as long-document question answering, summarization, and code completion. Complementing these efforts, ZeroSCROLLS (Shaham et al., 2023) and L-Eval (An et al., 2023) have broadened the scope to include a diverse array of practical natural language tasks, including query-driven summarization.

In parallel, retrieval benchmarks have largely relied on synthetic datasets, offering researchers precise control over variables such as the length of input tokens. This approach minimizes the impact of disparate parametric knowledge resulting from varied training methodologies. A significant body of recent work has concentrated on the development of synthetic tasks specifically for retrieval evaluation (Kamradt, 2023; Mohtashami & Jaggi, 2023; Li et al., 2023; Liu et al., 2024c; Hsieh et al., 2024). In addition, researchers have explored the potential of extended contexts in facilitating various forms of reasoning (Tay et al., 2021).

This dual focus on synthetic retrieval tasks and comprehensive understanding benchmarks reflects the field's commitment to rigorously assessing the capabilities of long-context models across diverse linguistic challenges.

**Prompting Compression**   In the field of prompt compression, various designs effectively combine semantic information to compress natural language. Wingate et al. (2022) utilize soft prompts to encode more information with fewer tokens. Chevalier et al. (2023) present AutoCompressor, which uses soft prompts to compress the input sequence and extend the original length of the base model. Both Zhou et al. (2023) and Wang et al. (2023) recurrently apply LLMs to summarize input texts, maintaining long short-term memory for specific purposes such as story writing and dialogue generation. The LLMLingua series (Jiang et al., 2023b; 2024; Fei et al., 2024) explores the potential of compressing LLM prompts in long-context, reasoning, and RAG scenarios. Fei et al. (2024) use pre-trained language models to chunk the long context and summarize semantic information, compressing the original context.

## D   STATISTICS OF MODELS

Table 19 provides configuration parameters for LLMs that we evaluated in our experiments.

| Model Name | LLaMA-3-8B-Instruct | Mistral-7B-Instruct-v0.2 & 0.3 | Qwen2-7B-Instruct |
|---|---|---|---|
| $L$ (Number of layers) | 32 | 32 | 28 |
| $N$ (Number of attention heads) | 32 | 32 | 28 |
| $D$ (Dimension of each head) | 128 | 128 | 128 |

Table 19: Models Configuration Parameters

## E STATISTICS OF DATASETS

Table 20 shows the statistics of the datasets that we used in our experiments.

| DATASET | # TRAIN | # TEST |
|---------|---------|--------|
| GSM8K (COBBE ET AL., 2021) | 7,473 | 1,319 |
| LONGBENCH (BAI ET AL., 2024) | - | 4,750 |
| NIAH* (KAMRADT, 2023) | - | 800 |

Table 20: Dataset Statistics. # TRAIN and # TEST represent the number of training and test samples, respectively. *: The size of the NIAH test set varies based on the context length and step size, typically around 800 samples per evaluation.

## F PROMPT

Table 21 shows the prompt for the Figure 1

| The prompt for demonstration |
|---|
| · · · · · · |
| · · · · · · |
| The purple-crested turaco (Gallirex porphyreolophus) or, in South Africa, the purple-crested loerie, (Khurukhuru in the Luvenḓa (Venḓa) language) is a species of bird in the clade Turaco with an unresolved phylogenetic placement. Initial analyses placed the purple-crested turaco in the family Musophagidae, but studies have indicated that these birds do not belong to this family and have been placed in the clade of Turacos with an unresolved phylogeny. It is the National Bird of the Kingdom of Eswatini, and the crimson flight feathers of this and related turaco species are important in the ceremonial regalia of the Swazi royal family. This bird has a purple-coloured crest above a green head, a red ring around their eyes, and a black bill. The neck and chest are green and brown. The rest of the body is purple, with red flight feathers. Purple-crested turacos are often seen near water sources, where they can be observed drinking and bathing, which helps them maintain their vibrant plumage. Purple-crested turacos are considered to be large frugivores that are known to carry cycad seeds from various plant species long distances from feeding to nesting sites. After fruit consumption, they regurgitate the seeds intact where they can germinate nearby. G. porphyreolophus primarily consumes fruits whole like many other large frugivores which are suggested to be necessary for effective ecosystem functioning. Among similar turacos, the purple-crested turaco have faster minimum transit times when consuming smaller seed diets than larger seed diets, and G. porphyreolophus has been shown to have significantly faster pulp (seedless fruit masses) transit time than another closely related Turaco when fed only the pulp of larger-seeding fruits than smaller-seeding fruits. In addition to their frugivorous diet, these birds are occasionally seen foraging for other food items such as nuts and leaves, which provide essential nutrients. They are also known to coexist with various other animals, including those that might enjoy strawberries and other similar fruits. The purple-crested turaco's role in seed dispersal is crucial, and their interaction with different elements of their habitat, including water and diverse plant materials, highlights their importance in maintaining ecological balance. |
| · · · · · · |
| · · · · · · |

Table 21: The prompt for demonstration

Here we provide the CoT prompt exemplars for GSM8K which is used in section 4.2.

## G LIMITATIONS

The major limitation of the ChunkKV is that it uses fixed-size token groups for chunking. While adaptive chunking methods could potentially improve performance, they would introduce significant inference latency. Therefore, finding a balance between the chunking method and inference latency is key to improving KV cache compression.

| GSM8K experiemnt CoT Prompt Exemplars |
|---|
| Question: There are 15 trees in the grove. Grove workers will plant trees in the grove today. After they are done, there will be 21 trees. How many trees did the grove workers plant today? |
| There are 15 trees originally. Then there were 21 trees after some more were planted. So there must have been 21 - 15 = 6. The answer is 6. |
| Question: If there are 3 cars in the parking lot and 2 more cars arrive, how many cars are in the parking lot? |
| There are originally 3 cars. 2 more cars arrive. 3 + 2 = 5. The answer is 5. |
| Question: Leah had 32 chocolates and her sister had 42. If they ate 35, how many pieces do they have left in total? |
| Originally, Leah had 32 chocolates. Her sister had 42. So in total they had 32 + 42 = 74. After eating 35, they had 74 - 35 = 39. The answer is 39. |
| Question: Jason had 20 lollipops. He gave Denny some lollipops. Now Jason has 12 lollipops. How many lollipops did Jason give to Denny? |
| Jason started with 20 lollipops. Then he had 12 after giving some to Denny. So he gave Denny 20 - 12 = 8. The answer is 8. |
| Question: Shawn has five toys. For Christmas, he got two toys each from his mom and dad. How many toys does he have now? |
| Shawn started with 5 toys. If he got 2 toys each from his mom and dad, then that is 4 more toys. 5 + 4 = 9. The answer is 9. |
| Question: There were nine computers in the server room. Five more computers were installed each day, from monday to thursday. How many computers are now in the server room? |
| There were originally 9 computers. For each of 4 days, 5 more computers were added. So 5 * 4 = 20 computers were added. 9 + 20 is 29. The answer is 29. |
| Question: Michael had 58 golf balls. On tuesday, he lost 23 golf balls. On wednesday, he lost 2 more. How many golf balls did he have at the end of wednesday? |
| Michael started with 58 golf balls. After losing 23 on tuesday, he had 58 - 23 = 35. After losing 2 more, he had 35 - 2 = 33 golf balls. The answer is 33. |
| Question: Olivia has $23. She bought five bagels for $3 each. How much money does she have left? |
| Olivia had 23 dollars. 5 bagels for 3 dollars each will be 5 x 3 = 15 dollars. So she has 23 - 15 dollars left. 23 - 15 is 8. The answer is 8. |

Table 22: GSM8K CoT Prompt Exemplars

# H LICENSES

For the evaluation dataset, all the datasets, including, GSM8K (Cobbe et al., 2021), LongBench (Bai et al., 2024) are released under MIT license. NIAH (Kamradt, 2023) is released under GPL-3.0 license.

