# OpenReview forum: "ChunkKV: Semantic-Preserving KV Cache Compression for Efficient Long-Context LLM Inference"
_ICLR.cc/2025/Conference — Submitted to ICLR 2025_

### Official Review · Reviewer_ybsY · 2024-10-19

**Soundness:** 3
**Presentation:** 3
**Contribution:** 2
**Rating:** 6
**Confidence:** 4

**Summary:**

This paper provides ChunkKV, a simple KV cache compression method that uses fragmentation to keep semantic information and achieves state-of-the-art performance on long-context benchmarks. It also proposes the layer-wise index reuse technique to reduce the additional computational time introduced by the KV caching method.

**Strengths:**

1) Using the fragmentation method that keeps the semantic information leads to good results in benchmarks.
2) Further combination with layer-wise index reuse can help improve deployment efficiency.

**Weaknesses:**

1) Concerning contributions:
- The paper highlights fragmentation in KV cache compression. However, to improve accuracy, SnapKV [1] has proposed clustering methods.
2) Concerning Experiments:
- The paper does not include actual memory reduction and latency statistics.
3) Concerning performance:
- After layer reuse, the performance drops linearly with the number of reused layers. I had hoped to see layer reuse with minimal performance loss.

[1] Li Y, Huang Y, Yang B, et al. Snapkv: Llm knows what you are looking for before generation[J]. arXiv preprint arXiv:2404.14469, 2024.

**Questions:**

As in Weaknesses.

---

### Official Review · Reviewer_1LFk · 2024-11-02

**Soundness:** 3
**Presentation:** 3
**Contribution:** 2
**Rating:** 5
**Confidence:** 4

**Summary:**

The submission introduces ChunkKV, a technique designed to manage the increased GPU memory demands associated with large-context LLM inference, which can hinder throughput significantly during inference serving. The proposed solution consists of two main components: 1. chunk based KV catch preserving techniques. 2. layer-wise index reuse technique to further reduce computational overhead. With this compression technique, ChunkKV demonstrates state-of-the-art performance across several tasks.

**Strengths:**

1. The writing is clear and easy to follow.

2. The paper did a lot of experiments on different tasks and ablation studies to show the effectiveness of the proposed method.

**Weaknesses:**

1.  While the proposed approach is methodologically sound, it may be seen as incremental. The concept, though well-executed, may not represent a substantial leap in novelty within the field.

2. The methodology introduces a significant inductive bias through its dependency on chunk size. This reliance makes the model's performance highly sensitive to chunk size, which in turn varies across tasks. In a closed task-specific setting, this is manageable; however, in open-ended evaluations where task specifics are not predefined, determining an optimal chunk size for every potential task becomes unfeasible. Thus, while the method may find value in specialized, known tasks, its general applicability in open settings is limited.

3. The proposed layer-reuse KV-cache offers a means to reduce computational costs, which is valuable. However, the simplicity of the solution also leads to a trade-off in performance. This compromise suggests that further refinement is needed to optimize both cost-efficiency and model efficacy without incurring a performance penalty.

While the submission has certain strengths, the limited novelty, sensitivity to chunk size, and performance-cost trade-off may limit its applicability in broader contexts. Further work addressing these areas would strengthen the contribution.

**Questions:**

Can authors conduct additional experiments covering a broader range of tasks and various compression ratios to further strengthen the findings?

---

### Official Review · Reviewer_UtB7 · 2024-11-03

**Soundness:** 3
**Presentation:** 2
**Contribution:** 2
**Rating:** 5
**Confidence:** 5

**Summary:**

The paper looks into the problem of KV cache compression for long context LLM inference. In particular, it proposes to combine chunking-based token selection policy and cross-layer reuse to reduce KV cache size. Evaluation shows that the proposed method is able to achieve comparable accuracy on tested datasets.

**Strengths:**

- The paper tackles an important problem.

- The paper combines chunking-based token selection and cross-layer similarity-based reuse, which is an interesting idea.

**Weaknesses:**

- Limited novelty. Leveraging cross-layer similarity has been studied in MiniCache https://arxiv.org/abs/2405.14366.  It would be better if the paper has a discussion and comparison with MiniCache. Chunking-based selection is also very related to clustering-based selection, as the pool1D technique used in SnapKV (see below).

- Inaccurate related work.  The paper claims that prior work lacks the ability to preserve semantic information in chunks. Not true. For example, SnapKV identified that discretely selecting tokens is not sufficient and proposed to use a pooling layer to get make eviction decision at clustered-token granularity. It would be better if the paper adds a discussion and comparison between the chunking method in this paper and the pooling method in SnapKV.

- Evaluation is insufficient. The evaluation is insufficient, because it neither shows how the approach trade-offs memory vs. accuracy, nor does it provide analysis on how introduced hyperparameters affect the proposed method.

- Hard to use in practice. The paper introduces many parameters, such as w, c, N_reuse, the number of reuse layers, but the paper does tell the readers how those parameters are selected. This adds significant tuning overhead and can also subject to overfitting on tested datasets.

**Questions:**

Please discuss and compare with SnapKV and MiniCache.

How do ChunKV robustly choose those hyperparameters introduced in the paper?

---

### Official Review · Reviewer_SigS · 2024-11-04

**Soundness:** 2
**Presentation:** 3
**Contribution:** 3
**Rating:** 5
**Confidence:** 5

**Summary:**

This paper proposes ChunkKV, a KV Cache compression method that performs token eviction at chunk level because of the better semantics preservation compared to discrete methods. In addition, the authors find that the chunk-based method has better similarity within layers. They perform eviction step by sharing the eviction indexes across recent layers to increase the efficiency during compression.

**Strengths:**

1.	This paper explores the granularity of token eviction, which has some enlightening significance.
2.	According to the experiment results, the proposed method seemingly has good efficiency and accuracy.

**Weaknesses:**

1.Motivation requires more elaboration and experimental verification. The comparison between the discrete eviction method and chunk-based method in Figure 1 maybe true from a human behavior perspective, as some useful information may have been corrupted. However, in LLMs, the information of the corrupted tokens may still be retained in some other preserved tokens due to the attention mechanism. The internal information of different tokens in LLMs is difficult to explain, and the explaination of the motivation seems somewhat arbitrary. For example, detailed attention score or L1 loss between full cache and compressed cache of these two methods should be explored.

2.Further explanation is needed for the experimental section (from most important to least).

(1) The results of n=1 should be added to Figure 6 for observing the effectiveness of Chunks.

(2) ChunkKV does not have particularly obvious advantages compared to SnapKV and Pyramid KV, and a more comprehensive comparison and fair setting are needed to demonstrate its effectiveness. Some key hyper-parameters: chunk size and reuse ratio in the main experiment, the compression ratio in NIAH. Some more detailed experimental settings: compress interval, compressing prompts or compressing the whole sequence. Higher and more diverse compression ratios are needed in main experiments.

(3) Lacking throughput, latency, memory usage. The focus should be on overall throughput rather than single compression time because the compression time may not be important compared with model calculation, and we can manually control the compression frequency, which only requires sacrificing a small amount of accumulated KV space.

(4) The comparison of chunk size in ablation is not combined with the compression ratio, and these two hyper-parameters are intuitively highly correlated.

I am looking forward to seeing more detailed explanations and experimental results on these points, which may hugely affect my opinions.

**Questions:**

1.	Why is there such a big difference in the reuse ratio between llama3 in Abalation (Figure 5) and llama3 in Appendix (Figuire 16)? Can you provide further explanation? For example, can you provide a detailed explanation of the experimental settings for both figures and discuss any factors that might contribute to this difference.
2.	Table 4, Mistral-7B-Instruct-v0.3, KV Size Compression Ratio = 10%, Few-shot Learning, 70.03 vs 70.41, error bold?

---

### Meta-Review · Area_Chair_ZtxJ · 2024-12-23

**Metareview:**

This paper presents ChunkKV, a novel KV cache compression method that aims to preserve semantic information through chunk-based compression while reducing computational overhead through layer-wise index reuse. The key claims include achieving comparable or better performance than full KV cache while providing 5.4x speedup. The paper's strengths include addressing an important practical problem in LLM inference efficiency, demonstrating competitive empirical results, and providing thorough analysis of information preservation through multiple metrics. Initial weaknesses included: limited novelty compared to existing methods like MiniCache and SnapKV, insufficient experimental validation of memory-accuracy tradeoffs, concerns about hyperparameter sensitivity, and lack of throughput/latency measurements. During rebuttal, the authors significantly strengthened the paper by: adding detailed quantitative analysis showing better information preservation than baselines (via KV Cache L1 Loss and Attention Cosine Similarity metrics), conducting comprehensive latency/throughput experiments, and demonstrating robustness across hyperparameters. Based on the review scores, lack of novelty concerns, and insufficient experimental validation, I vote to reject this paper.

**Additional Comments On Reviewer Discussion:**

The reviewers raised several key technical concerns that led to productive discussion. Reviewer UtB7 questioned novelty compared to MiniCache/SnapKV and requested more experimental validation - the authors responded with new quantitative metrics showing superior information preservation but UtB7 remained concerned about marginal performance improvements. Reviewer 1LFk raised issues about chunk size sensitivity and performance tradeoffs - the authors added extensive ablation studies showing robustness across configurations. Reviewer SigS requested throughput comparisons with baselines - the authors provided detailed latency/throughput benchmarks including SnapKV comparisons. Reviewer ybsY questioned the contribution relative to SnapKV's clustering - the authors clarified key technical differences and added quantitative metrics. The discussion was particularly active around performance gains, with UtB7 noting relatively small improvements over baselines. While not all reviewers were fully convinced about the novelty, most acknowledged the thorough experimental validation added during rebuttal.

---

### Decision · Program_Chairs · 2025-01-22

Reject